# Amplified Patch-Level Differential Privacy for Free via Random Cropping

**Kaan Durmaz**                                        *k.durmaz@tum.de*
*Technical University of Munich*

**Jan Schuchardt**                        *jan.a.schuchardt@morganstanley.com*
*Machine Learning Research, Morgan Stanley*
*Technical University of Munich*

**Sebastian Schmidt**                              *sebastian95.schmidt@tum.de*
*Technical University of Munich*

**Stephan Günnemann**                                  *s.guennemann@tum.de*
*Technical University of Munich*

**Reviewed on OpenReview:** *https://openreview.net/forum?id=pSWuUF8AVP*

## Abstract

Random cropping is one of the most common data augmentation techniques in computer vision, yet the role of its inherent randomness in training differentially private machine learning models has thus far gone unexplored. We observe that when sensitive content in an image is spatially localized, such as a face or license plate, random cropping can probabilistically exclude that content from the model's input. This introduces a third source of stochasticity in differentially private training with stochastic gradient descent, in addition to gradient noise and minibatch sampling. This additional randomness amplifies differential privacy without requiring changes to model architecture or training procedure. We formalize this effect by introducing a patch-level neighboring relation for vision data and deriving tight privacy bounds for differentially private stochastic gradient descent (DP-SGD) when combined with random cropping. Our analysis quantifies the patch inclusion probability and shows how it composes with minibatch sampling to yield a lower effective sampling rate. Empirically, we validate that patch-level amplification improves the privacy-utility trade-off across multiple segmentation architectures and datasets. Our results demonstrate that aligning privacy accounting with domain structure and additional existing sources of randomness can yield stronger guarantees at no additional cost.[1]

## 1 Introduction

Differential Privacy (DP) (Dwork, 2006; Dwork and Roth, 2014) provides a mathematically rigorous paradigm for privacy protection. It limits the influence of any individual data point on the output of a learning algorithm and offers provable safeguards against privacy attacks. For instance, it protects against membership inference (Shokri et al., 2017), where an adversary attempts to determine whether a particular data point was part of the training set, and model inversion, where an adversary tries to reconstruct sensitive features from model outputs (e.g. faces from a facial recognition algorithm (Fredrikson et al., 2015)). This guarantee holds regardless of the adversary's prior knowledge. It is formalized through a neighboring relation, typically defined as two datasets differing in a single data point, which is denoted as $x \simeq x'$.

---

[1]Full implementation details are provided in `github.com/TUM-DAML/patch_level_dp`.

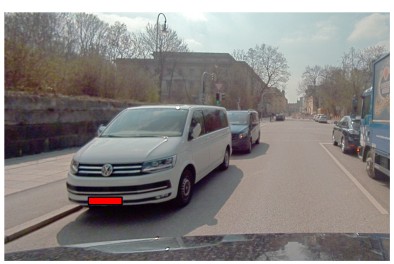 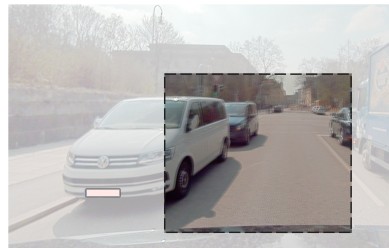 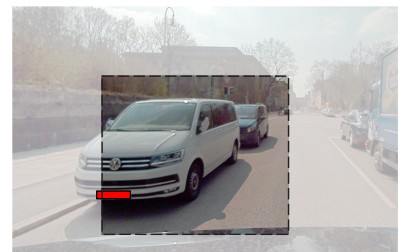

🟥 Private Patch    ⬜ Random Crop

Figure 1: Illustration of the effect of random cropping on a private patch as the license plate (in red). Left: Original image with a designated private patch. Middle: A random crop that excludes the private patch. Right: A random crop that includes a part of the private patch. Any intersection allows the private patch to influence the output of the model.

In deep learning, the most widely used method for achieving DP is DP-SGD (Song et al., 2013; Abadi et al., 2016). It is an elegant and simple modification of SGD, which works by adding noise to the clipped per-sample gradients. A key aspect of DP-SGD is amplification by subsampling, where privacy is enhanced by applying a differentially private mechanism to randomly selected batches from the dataset (Kasiviswanathan et al., 2008; Li et al., 2011). The idea behind this is that the private data contributes to each training step with only some small probability. Thus, the mechanism reveals less information about any single individual on average, and the overall privacy cost is reduced through structured randomness.

DP-SGD is broadly applicable across domains where datasets can be represented as sets of arbitrary records, such as images (Abadi et al., 2016) or texts (Anil et al., 2022). However, this generality also presents a limitation: it does not take advantage of domain-specific structure. In particular, stronger privacy guarantees may be possible when we (1) have additional knowledge about a domain, (2) can make more concrete assumptions about what constitutes private information, and (3) are able to redefine the neighboring relation to better match the nature of privacy risks in that domain. This opens the door to more refined and tighter privacy mechanisms that are still formally sound, but better aligned with how sensitive information manifests and is structured across different domains.

Building on this motivation, in this work we take advantage of domain-specific structure in vision. In many real-world vision applications, privacy-sensitive content is not spread across the entire image, but instead localized to small, well-defined *patches*: a face in a photo, a license plate in a traffic scene. In such cases, assuming that the entire image is private is unnecessarily conservative. Instead, it is more natural to define privacy at the patch level. This motivates a patch-level neighboring relation, where datasets differ by a small modification inside one image, leaving the rest untouched.

Under this proposed patch-level notion of privacy, we investigate whether the privacy guarantees of DP-SGD can be improved. Our central observation is that we can leverage an already standard component of most vision training pipelines, *random cropping*. Cropping selects a random subregion or *crop* of each image in a minibatch during training and is commonly used to improve generalization and reduce compute in high-resolution tasks (Krizhevsky et al., 2012; Cordts et al., 2016; Chen et al., 2017; 2018; Shorten and Khoshgoftaar, 2019). When private information is spatially localized, cropping introduces a powerful, implicit form of randomness which can *exclude the sensitive region entirely* from the crop. This implicit randomness of cropping introduces stochasticity at the patch level within each image, analogous to the stochasticity at the dataset level introduced by minibatch sampling.

In this work, we formalize the analogy between cropping and minibatch sampling to rigorously analyze the resulting mechanism. Specifically, we:

- introduce a spatially localized, patch-level neighboring relation tailored to privacy in vision data,
- formalize random cropping as a privacy amplification mechanism and show that it probabilistically excludes sensitive regions,

- provide a tight theoretical privacy analysis under this new neighboring relation, demonstrating reduced sensitivity and improved privacy bounds for DP-SGD.

Importantly, our approach requires no changes to the training algorithm and introduces no additional computational overhead, making it a drop-in improvement in the privacy accounting of DP-SGD. The only assumption we make is that private information is spatially localized, an intuitive and realistic condition in many practical vision tasks.

We empirically validate our findings on semantic segmentation with DeepLabV3+ (Chen et al., 2018) and PSPNet (Zhao et al., 2017) trained on Cityscapes (Cordts et al., 2016) and A2D2 (Geyer et al., 2020), demonstrating that random cropping improves the privacy-utility trade-off in realistic training scenarios. We view this work as a step toward harnessing more sources of randomness already present in machine learning pipelines to achieve stronger privacy guarantees.

## 2 Related Work

**Privacy in Computer Vision.** A large body of prior work addresses the problem of releasing images in a privacy-preserving manner, i.e., obfuscating private information directly in images. Classical approaches such as blurring, pixelation, or mosaicing are intuitive but offer no protection against modern deep learning based attacks, which can often recover obfuscated content (McPherson et al., 2016; Oh et al., 2016). To address such model-based threats, some recent works explore GAN-based anonymization (Sun et al., 2017; Wu et al., 2019). Another direction seeks to minimize data exposure via encrypted processing (Dowlin et al., 2016). While effective for specific release scenarios, these approaches focus on privatizing the output images rather than the models trained on them.

**Differential Privacy for Vision Models.** DP-SGD (Abadi et al., 2016) is the standard approach for training differentially private models. However, recent works highlight that maintaining utility at scale requires extensive hyperparameter tuning and compute, but also places increasing importance on strong data augmentation to stabilize training and improve generalization (Ponomareva et al., 2023; Sander et al., 2022; De et al., 2022). In response to these challenges, some alternative approaches emerged that modify the training process, model architecture, or neighboring relation. AdaMix (Golatkar et al., 2022) improves utility in mixed differential privacy by using a small amount of task-aligned public data for adaptive model initialization and training. However, its effectiveness depends on the availability of labeled public data closely matching the target distribution. DP-Image (Liu et al., 2021) achieves differential privacy by adding noise to encoder outputs and reconstructing samples with a denoising GAN. However, the method is geared toward releasing visually acceptable images and is not evaluated against standard DP training methods like DP-SGD, leaving open questions about its utility for downstream learning. PixelDP (Lécuyer et al., 2018) connects adversarial robustness and differential privacy by enforcing guarantees against input perturbations at inference. While one can view this as a form of fine-grained privacy, its objective is fundamentally different: They focus on robustness against perturbations during inference, whereas we focus on privacy during training. Unlike all of the previously discussed methods, which require architectural changes or public data, we retain the standard DP-SGD training algorithm but improve its privacy analysis by accounting for the inherent randomness of commonly used data augmentations.

**Privacy Amplification via Structured Randomness.** Privacy amplification by subsampling is a core principle in differential privacy and is already leveraged in DP-SGD through minibatch sampling. Building on this, a few works have explored how other types of structured randomness in training can further improve privacy. Dong et al. (2025) focus on partial participation mechanisms. A key example they discuss is dropout, which introduces randomness over model parameters and thus amplifies privacy. However, De et al. (2022) found that parameter-level regularization techniques such as weight decay and dropout harm both training and validation performance in DP settings. In contrast, our work focuses on input-level augmentation. Schuchardt et al. (2025) show that combining minibatch sampling with additive Gaussian data augmentation can amplify privacy in time series forecasting. Their analysis proves that the Gaussian noise has a similar effect to subsampling in amplifying privacy. This amplification-by-augmentation result is derived for relaxed neighboring relation that bounds both the number and magnitude of changes to sequences. Our work focuses

on a different data domain (images), a different form of data augmentation (cropping), and does not make any assumptions about the magnitude of change. Similar to Lécuyer et al. (2018), Lin et al. (2021) provide a formal analysis of random cropping as a defense against adversarial attacks, establishing a connection between adversarial robustness and differential privacy. However, their analysis focuses on inference-time robustness and privacy, whereas we study training-time privacy guarantees through DP-SGD.

## 3 Background and Preliminaries

**Differential Privacy.** Differential Privacy (DP) (Dwork, 2006) formalizes the requirement that the output of a randomized algorithm remains nearly unchanged when a single individual's data is modified. Let $\mathcal{X}$ denote the space of datasets. A randomized mechanism $M : \mathcal{X} \to \mathbb{R}^D$ is said to satisfy DP if the distributions of $M(x)$ and $M(x')$ are almost indistinguishable whenever $x$ and $x'$ are *neighboring datasets* (denoted $x \simeq x'$), meaning they differ in the data associated with a single individual. Formally:

**Definition 1.** *A mechanism $M : \mathcal{X} \to \mathbb{R}^D$ satisfies $(\varepsilon, \delta)$-DP if for all measurable subsets $O \subseteq \mathbb{R}^D$ and for all $x, x' \in \mathcal{X}$ such that $x \simeq x'$, $\Pr[M(x) \in O] \leq e^\varepsilon \Pr[M(x') \in O] + \delta$.*

This condition can equivalently be expressed in terms of the hockey-stick divergence between $M(x)$ and $M(x')$:

**Proposition 1.** *(Barthe and Olmedo, 2013) Let $H_\alpha(P \| Q) := \int_{\mathbb{R}^D} \max\left\{\frac{dP}{dQ}(o) - \alpha, 0\right\} dQ(o)$. Then $M$ is $(\varepsilon, \delta)$-DP if and only if $H_{e^\varepsilon}(M(x) \| M(x')) \leq \delta$.*

As Proposition 1 suggests, and following the literature (e.g., Barthe and Olmedo (2013) and Zhu et al. (2022)), we use $\alpha$ and $e^\varepsilon$ interchangeably throughout the manuscript.

**Neighboring Dataset Relations.** The choice of neighboring relation $x \simeq x'$ depends on the application. Two standard variants are: The *insertion/removal relation*, denoted $x \simeq_\pm x'$, which holds when $x'$ is obtained by adding or removing a single element: $x' = x \cup \{a\}$ or $x' = x \setminus \{a\}$ for some element $a$. The *substitution relation*, denoted $x \simeq_\Delta x'$, which holds when $x$ and $x'$ have the same size and differ in one entry: $x' = x \setminus \{a\} \cup \{a'\}$ for some $a, a'$.

**Subsampling.** We consider a random subsampling scheme $S : \mathcal{X} \to \mathcal{Z}$ and an $(\varepsilon', \delta')$-DP base mechanism $B : \mathcal{Z} \to \mathbb{R}^D$. The subsampled mechanism $M = B \circ S$ first draws a random batch from the dataset and then applies the base mechanism to this batch.

**Proposition 2.** *Let $\mathcal{X}_n \subseteq \mathcal{X}$ denote the set of datasets of size $n$. For $m \leq n$, define the subsampling without replacement mechanism $S_m^{\mathrm{wo}} : \mathcal{X}_n \to \mathcal{Z}$ such that, given a dataset $x \in \mathcal{X}_n$, the mechanism $S_m^{\mathrm{wo}}(x)$ outputs a subset $y \subseteq x$ of size $|y| = m$ drawn uniformly at random (i.e. $\mathcal{Y} := \mathcal{X}_m$). Then the sampling rate is $\gamma_{wo} = m/n$.*

While our methods apply to arbitrary subsampling schemes, in practice we focus on subsampling without replacement as in Proposition 2. The following standard result makes explicit how the sampling rate $\gamma$ of such a scheme enters the privacy guarantees.

**Proposition 3.** *(Ullman, 2017; Balle et al., 2018) If $M'$ is an $(\varepsilon', \delta')$-DP randomized mechanism, then $M = M' \circ S$ obeys $(\varepsilon, \delta)$-DP with $\varepsilon = \log\left(1 + \gamma\left(e^{\varepsilon'} - 1\right)\right)$, $\delta = \gamma \delta'$, where $\gamma$ is the sampling rate of the subsampling scheme.*

**Dominating Pairs.** In DP-SGD (Song et al., 2013; Abadi et al., 2016) the mechanism $M$ is a single training step. Therefore, during a training run, we apply $M$ repeatedly. To determine the resulting $(\varepsilon, \delta)$ different *privacy accountants* can be used (Koskela et al., 2020; Meiser and Mohammadi, 2018; Dong et al., 2022; Sommer et al., 2019). Recently, Zhu et al. (2022) introduced the notion of *dominating pairs*, which fully characterizes the tradeoff between $\varepsilon$ and $\delta$, providing a unifying framework for tracking the privacy loss of composed mechanisms.

**Definition 2.** *Let $M$ be a randomized mechanism. A pair of distributions $(P, Q)$ with densities $(p, q)$ is said to be a dominating pair for $M$ iff $H_\alpha(M(x) \| M(x')) \leq H_\alpha(P \| Q)$ for all $x \simeq x'$ and all $\alpha \geq 0$, where $H_\alpha$ denotes the hockey-stick divergence.*

**Proposition 4.** *(Zhu et al., 2022) For the Gaussian mechanism, $(P, Q)$ with $P : \mathcal{N}(1, \sigma^2), Q : \mathcal{N}(0, \sigma^2)$ is a dominating pair.*

While Proposition 3 provides a simple and intuitive privacy amplification bound for $\varepsilon > 0$ (i.e., $\alpha > 1$), it does not cover the complementary regime $\varepsilon \leq 0$ (i.e., $0 < \alpha \leq 1$). To address this gap, we turn to the notion of dominating pairs, which yields a more general subsampling bound valid across the entire range of $\varepsilon$.

**Proposition 5.** *(Zhu et al., 2022) Let $\mathcal{M}$ be a mechanism with dominating pair $(P, Q)$ under the substitution relation $\simeq_\Delta$ and $S_m^{\mathrm{wo}}$ the subsampling mechanism from Proposition 2 with sampling rate $\gamma_{wo} = m/n$. Then the composed mechanism $\mathcal{M} \circ S_m^{\mathrm{wo}} : \mathcal{X}_n \to \mathbb{R}^D$ satisfies:*

$$\delta_{\mathcal{M} \circ S_m^{\mathrm{wo}}}(\alpha) \leq \begin{cases} H_\alpha((1 - \gamma_{wo})Q + \gamma_{wo}P \,\|\, Q) & \text{for } \alpha \geq 1, \\ H_\alpha(P \,\|\, (1 - \gamma_{wo})P + \gamma_{wo}Q) & \text{for } 0 < \alpha < 1. \end{cases}$$

This result provides a tight upper bound on the privacy profile of the subsampled mechanism. We will later improve upon this bound by leveraging the interaction of random cropping and patch-level privacy.

## 4 Privacy Amplification via Random Cropping

We study a form of privacy amplification that arises uniquely in vision models trained with data augmentations. Specifically, we show that the spatial structure of visual data, combined with random cropping, naturally gives rise to an amplification effect akin to that provided by minibatch subsampling in DP-SGD. Our analysis formalizes this intuition through the notion of *patch-level differential privacy.*

This section introduces the key components of our framework. First, we define a patch-level neighboring relation that captures localized substitutions within an image. Second, we model random cropping as a stochastic transformation over padded images. Finally, we characterize how often a fixed region of interest appears in a random crop, and use this to derive privacy bounds under patch-level substitutions.

### 4.1 Patch-Level Neighboring Relation

As previously introduced, differential privacy is defined with respect to a neighboring relation, which specifies when two datasets are considered adjacent. To capture localized privacy risks in vision tasks, we define a neighboring relation for bounded spatial regions in a single image. Specifically, we define it as the substitution of a rectangular patch. Substitution is a natural choice because insertion or removal would alter the size or structure of an image (incompatible with fixed-shape learning pipelines), while substitution preserves structure and realistically captures localized, contiguous changes of data.

**Definition 3.** *Let $\mathcal{I} := [0, 255]^{3 \times H_I \times W_I}$ denote the set of input images with $C = 3$ channels and spatial dimensions $H_I \times W_I$. Define $\mathcal{X}$ as the space of datasets of fixed size $n$, where each dataset $x = \{x_1, \ldots, x_n\} \in \mathcal{X}$ consists of $n$ images $x_i \in \mathcal{I}$. Let $R \subseteq [H_I] \times [W_I]$ be a fixed rectangular region of pixel coordinates (i.e., a patch). We write $x_i(s, t)$ for the pixel value at coordinates $(s, t)$ in image $x_i$ of dataset $x$. We say that two datasets $x, x' \in \mathcal{X}$ are patch-level substitution neighbors, denoted $x \simeq_{\Delta, p} x'$, if there exists an index $i \in [n]$ and a region $R$ such that $x_j = x'_j$ for all $j \neq i$, and $x_i(s, t) = x'_i(s, t)$ for all pixels $(s, t) \notin R$, but $x_i(s, t) \neq x'_i(s, t)$ for some pixels $(s, t) \in R$.*

In other words, the datasets differ only in a single spatially localized region $R$ within a single image, and are otherwise identical. Notably, even a single-pixel change within $R$ suffices to constitute adjacency. This formulation aligns with many real-world privacy scenarios in vision, where sensitive content is confined to a localized area, as discussed in the introduction, and serves as the foundation for our analysis of spatially-aware privacy mechanisms. At its core, this notion is analogous to the usage of record-level DP (protecting individual data points) (Dwork et al., 2006; Dwork, 2006), compared to user-level DP (protecting all data from a single user) (Levy et al., 2021). Coarser relations are always valid, but finer-grained relations yield tighter guarantees when domain structure permits. We provide a detailed discussion of when patch-level privacy is appropriate versus record-level privacy in Appendix A.

## 4.2 Random Cropping as a Stochastic Mechanism

Random cropping introduces stochasticity at the spatial level by selecting subregions of an image. It is primarily used as a data augmentation technique to improve generalization (Krizhevsky et al., 2012; Shorten and Khoshgoftaar, 2019), but this randomness can also be interpreted as a stochastic transformation, akin to minibatch sampling in DP-SGD. To formalize random cropping, we first account for the fact that, in practice, images could be symmetrically padded before cropping. Padding enlarges the spatial domain without altering the image content, and all subsequent operations are then performed on the padded image. Given an image $x_i \in \mathcal{I}$, we use $\tilde{x}_i \in [0, 255]^{3 \times (H_I + 2\mathrm{pad}_y) \times (W_I + 2\mathrm{pad}_x)}$ to denote the corresponding symmetrically padded image, where $\mathrm{pad}_x, \mathrm{pad}_y \in \mathbb{N}_0$ are the horizontal and vertical padding sizes. In the following, all coordinates are defined with respect to the padded image, using a bottom-left origin.

**Definition 4.** *Let $H_C \times W_C$ be a fixed crop size. The set of valid crop origins is $\Omega := \left\{ (u, v) \in \mathbb{N}_0^2 \,\middle|\, 0 \le u \le W_I + 2\mathrm{pad}_x - W_C, \quad 0 \le v \le H_I + 2\mathrm{pad}_y - H_C \right\}$. For each origin $(u, v) \in \Omega$, the corresponding crop region is $C_{u,v} := \left\{ (i, j) \in \mathbb{N}_0^2 \,\middle|\, u \le i < u + W_C, \quad v \le j < v + H_C \right\}$.*

In words, Definition 4 specifies the set of all possible rectangular windows of size $H_C \times W_C$ that can be extracted from the padded image.

**Definition 5.** *The random cropping mechanism $S^{\mathrm{crop}} : \mathcal{I} \to \mathcal{I}_C$ samples a crop origin $(u, v) \sim \mathrm{Unif}(\Omega)$ from the set of valid origins (Definition 4) and returns*

$$S^{\mathrm{crop}}(x_i) := \tilde{x}_i\big|_{C_{u,v}}, \tag{1}$$

*where $\tilde{x}_i$ is the padded image and $\tilde{x}_i\big|_{C_{u,v}}$ denotes its restriction to the crop region $C_{u,v}$.*

## 4.3 Patch Inclusion Probability

To analyze privacy amplification under patch-level substitution, we first need to quantify how likely it is that a sensitive region is included in a crop. If a private patch is excluded entirely, the mechanism's output becomes independent of that region, meaning it cannot influence the gradient. Conversely, any intersection means the private region can influence the output. This motivates the definition of the *patch inclusion probability*, i.e., the probability that a randomly chosen crop intersects with region $R$.

**Definition 6.** *Let $R \subseteq \mathbb{N}_0^2$ be a fixed rectangular region in the padded image. Let $\Omega_R := \{ (u, v) \in \Omega \mid C_{u,v} \cap R \ne \emptyset \}$ be the set of crop origins for which the corresponding crop region $C_{u,v}$ intersects $R$. Then the patch-level inclusion probability is $\gamma'_{\mathrm{crop}}(R) := \frac{|\Omega_R|}{|\Omega|}$.*

Intuitively, this probability increases with crop size and decreases with image size. Under the neighboring relation $\simeq_{\Delta,p}$, two datasets differ only in a single spatially localized region $R$. In analogy to standard amplification by subsampling, where each record is included with some probability, we treat $\gamma'_{\mathrm{crop}}$ as the sampling rate at which the private region is included in the mechanism's input. We now derive a closed-form expression for this quantity, assuming a fixed patch location.

**Lemma 1.** *Let $R \subseteq \mathbb{N}_0^2$ be a fixed rectangular private patch in the* original *image, with bottom-left coordinate $(R_x, R_y)$ and size $H_R \times W_R$. After applying symmetric padding of $\mathrm{pad}_x$ pixels horizontally and $\mathrm{pad}_y$ pixels vertically, we denote its bottom-left coordinate in the* padded *image by $R'_x = R_x + \mathrm{pad}_x$, $R'_y = R_y + \mathrm{pad}_y$. Let $(X_{\min}, Y_{\min})$ and $(X_{\max}, Y_{\max})$ denote the minimum and maximum crop origins for which the crop region $C_{u,v}$ intersects $R$. These are given by:*

$$X_{\min} = \max\left(0,\, R'_x - W_C + 1\right), \qquad X_{\max} = \min\left(W_I + 2\mathrm{pad}_x - W_C,\, R'_x + W_R - 1\right),$$
$$Y_{\min} = \max\left(0,\, R'_y - H_C + 1\right), \qquad Y_{\max} = \min\left(H_I + 2\mathrm{pad}_y - H_C,\, R'_y + H_R - 1\right).$$

*Then, the* patch-level inclusion probability *(Definition 6) is:*

$$\gamma'_{\mathrm{crop}}(R) = \frac{(X_{\max} - X_{\min} + 1)(Y_{\max} - Y_{\min} + 1)}{(W_I + 2\,\mathrm{pad}_x - W_C + 1)(H_I + 2\,\mathrm{pad}_y - H_C + 1)}. \tag{2}$$

*Proof.* See Appendix B.1.

### 4.4 Worst-Case Private Patch Exposure

The definition of $\gamma'_{\text{crop}}(R)$ above provides the inclusion probability for a fixed patch location $R$. For sound privacy guarantees, however, we need to account for the worst-case placement of the private region, i.e., the position that maximizes its inclusion probability and thus its potential privacy exposure.

**Definition 7.** *The worst-case patch inclusion probability is defined as $\gamma_{\text{crop}} := \max_{R} \gamma'_{\text{crop}}(R)$, where the maximization ranges over all valid placements of $R$ within the padded image domain.*

**Theorem 1.** *Let $R_{center} = \left( \left\lfloor \frac{H_I - H_R}{2} \right\rfloor, \left\lfloor \frac{W_I - W_R}{2} \right\rfloor \right)$ denote the centrally placed patch. Then, the worst-case patch inclusion probability is achieved at $R_{center}$, i.e., $\arg\max_{R} \gamma'_{\text{crop}}(R) = R_{center}$.*

*Proof.* See Appendix B.2.

The patch inclusion probability quantifies an image-level stochasticity that is distinct from, and composes naturally with, the minibatch-level sampling performed in DP-SGD. Specifically, each image undergoes two layers of randomness: first, random cropping at the patch level, and second, dataset-level minibatch subsampling without replacement.

This leads to an effective sampling rate $\gamma_{\text{eff}} := \gamma_{\text{wo}} \cdot \gamma_{\text{crop}}$, where $\gamma_{\text{wo}} = m/n$ is the minibatch sampling rate and $\gamma_{\text{crop}}$ is the worst-case patch inclusion probability. We now state a result that makes this interaction explicit in terms of differential privacy.

**Theorem 2.** *Let $\mathcal{M}$ be a mechanism that is dominated by $P, Q$ under the substitution relation $\simeq_{\Delta}$. Let $S_m^{\text{wo}}$ denote without-replacement subsampling of minibatch size $m$ and $S^{\text{crop}}$ the random cropping mechanism we defined in Definition 5. Then, under the patch-level substitution relation $\simeq_{\Delta,p}$, the composed mechanism $\mathcal{M} \circ S^{\text{crop}} \circ S_m^{\text{wo}}$ satisfies*

$$\delta_{\mathcal{M} \circ S^{\text{crop}} \circ S_m^{\text{wo}}}(\alpha) \leq \begin{cases} H_\alpha \left( (1 - \gamma_{\text{eff}})Q + \gamma_{\text{eff}}P \,\|\, Q \right) & \text{if } \alpha \geq 1, \\ H_\alpha \left( P \,\|\, (1 - \gamma_{\text{eff}})P + \gamma_{\text{eff}}Q \right) & \text{if } 0 < \alpha < 1, \end{cases} \tag{3}$$

*where $\gamma_{\text{eff}} := \gamma \cdot \gamma_{\text{crop}}$, with minibatch sampling rate $\gamma = \frac{m}{n}$ and worst-case patch inclusion probability $\gamma_{\text{crop}}$.*

*Proof.* See Appendix B.3.

This theorem captures the central insight of our work. It shows that the randomness already present in data augmentation, i.e., random cropping, can be leveraged as an additional source of privacy amplification *for free*. This amplification is possible because we adopt a spatially localized neighboring relation, which aligns the privacy definition with the structure of vision data. Together, these elements explain why, for any minibatch sampling rate $\gamma$, our privacy guarantees are strictly stronger than the usual tight analysis of DP-SGD from Proposition 5 would suggest. Not only are the guarantees strictly stronger, but one can also control both sources of randomness independently to attain a wider range of privacy-utility trade-offs.

**Tightness for DP-SGD.** Eq.3 provides a sound upper bound for arbitrary mechanisms. In B.4, we additionally prove that the bound holds with equality when $P, Q$ are a tight dominating pair of the mechanism $\mathcal{M} : 2^{\mathcal{I}_C} :\to \mathbb{R}^D$ that maps batches of cropped images to clipped, noised, and averaged gradients for some worst-case model $f$. Without further assumptions about the model, it is therefore the strongest possible privacy guaranty that can be derived for DP-SGD with patch-level subsampling.

**Generalization to Arbitrary Regions.** While we assume rectangular patches here for clarity and tractability, the same reasoning applies to other shapes as well. For certain simple geometries such as circular regions (e.g., faces), it may still be possible to derive closed-form inclusion probabilities. For irregular blobs, however, no compact expression exists, and the probability must instead be calculated using computational methods. The more faithfully the private region is modeled by its true structure, the less conservative the assumptions need to be, resulting in tighter inclusion probability bounds and correspondingly stronger privacy amplification from domain-specific knowledge. We demonstrate this empirically with varied geometries in Appendix D.1.6.

**Generalization to Multiple Patches.** Our analysis extends naturally to images containing multiple disjoint sensitive regions. One could (1) use the group privacy property of DP (Vadhan, 2017), or (2) assume

the union of these patches as a single private blob and directly use our tight analysis as discussed above. In either case, the worst-case limit of multiple patches covering the entire image recovers standard DP-SGD.

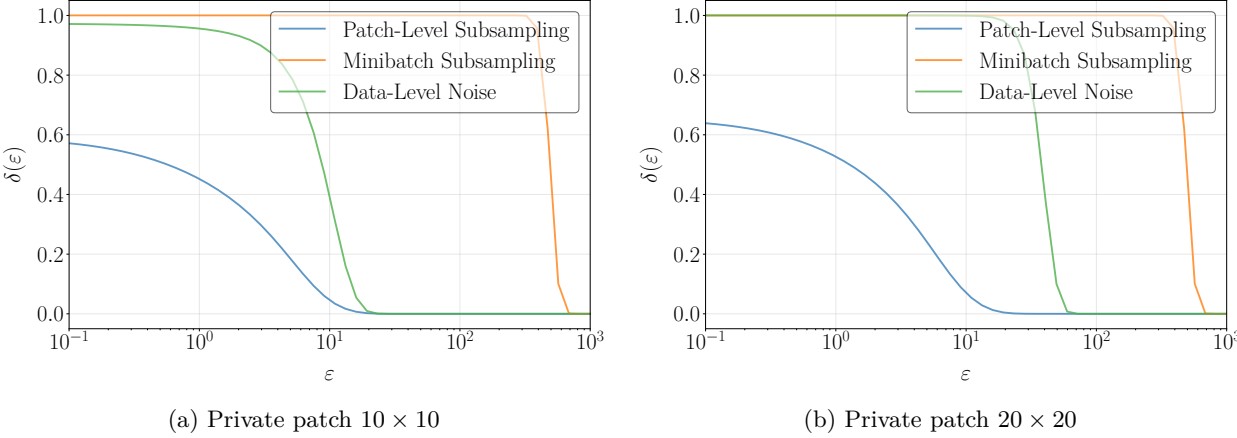

(a) Private patch $10 \times 10$          (b) Private patch $20 \times 20$

Figure 2: Privacy profiles for different mechanisms. DP-SGD with patch-level subsampling (blue) achieves stronger privacy amplification compared to standard minibatch subsampling (orange) with identical $\sigma = 1$. Even with an an extremely large noise scale $\sigma_{\text{data}} = 1000$, data-level noise addition (green) is less private.

## 5 Experimental Evaluation

We now empirically investigate the impact of patch-level subsampling on both privacy guarantees and model utility. Our goal is to understand how patch-level differential privacy interacts with key parameters and to validate that patch-level amplification provides tangible advantages over standard subsampling strategies. When we refer to patch-level subsampling, we mean this mechanism applied on top of standard minibatch subsampling. We organize our results into three parts. (1) Direct comparison of privacy profiles across different sampling strategies, (2) dependence of privacy amplification on crop and patch size, and (3) privacy-utility tradeoffs under patch-level subsampling versus standard minibatch subsampling.

### 5.1 Mechanism-Specific Privacy Profiles

To begin, we compare the privacy of three mechanisms in isolation: standard DP-SGD with minibatch subsampling, our patch-level sampling combined with minibatch subsampling, and a baseline where Gaussian noise is added directly to the input images. This highlights the privacy properties of patch-level amplification independently of downstream training performance.

For all methods, we compute the privacy profile $\delta(\varepsilon)$ as a function of $\varepsilon$. We use a fixed setup with crop size $100 \times 100$, private patch sizes $10 \times 10$ (Figure 2a) and $20 \times 20$ (Figure 2b). For the two subsampling methods, we assume a Gaussian noise multiplier $\sigma = 1$. Since the privacy profiles are sensitive to many hyperparameters (as discussed in (Ponomareva et al., 2023; Sander et al., 2022; De et al., 2022)), we report results here for this representative configuration and defer additional combinations to Appendix D.1.

The data-level noise addition baseline is included to address a natural question: can a simpler alternative, such as directly adding Gaussian noise to the input pixels before training, provide comparable privacy guarantees? We scale the noise using the expected sensitivity of a *localized private patch*, parallel to our method. In particular, we compute the $\ell_2$-sensitivity as $\Delta_2 := 255 \cdot \sqrt{3 \cdot H_R \cdot W_R}$, where $H_R, W_R$ are the side lengths of the private region.

Figure 2 shows the resulting privacy profiles. Patch-level sampling (blue) consistently achieves lower privacy leakage across a wide range of $\varepsilon$ values compared to standard DP-SGD (orange), confirming that incorporating spatial structure into the mechanism leads to stronger privacy guarantees. It also strictly outperforms the noise addition baseline (green): even with a noise standard deviation as large as $\sigma_{\text{data}} = 1000$, privacy

is uniformly worse than with patch-level subsampling. This extreme noise level further exceeds the dynamic range of non-normalized image pixels $[0, 255]$, underscoring the impracticality of uniform noise injection as a DP strategy for visual data.

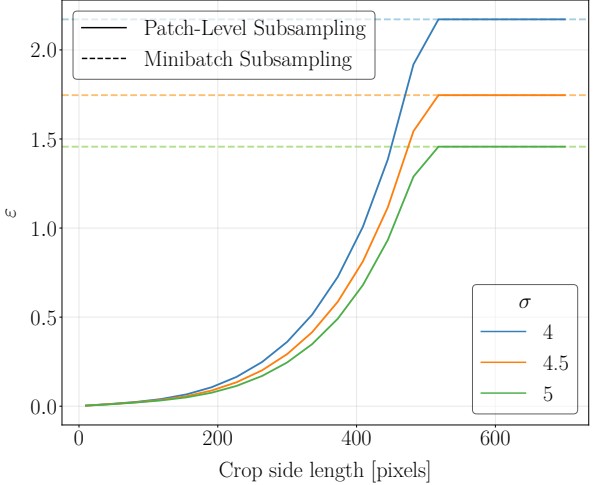

Figure 3: Varying crop size for patch-level subsampling, for different noise levels and $\delta = 10^{-5}$. The privacy parameter $\varepsilon$ increases rapidly with the crop size. All curves saturate at the minibatch subsampling baseline once the intersection probability becomes 1.

Figure 4: Varying private patch size for patch-level subsampling, for different noise levels and $\delta = 10^{-5}$. The privacy parameter $\varepsilon$ increases approximately linearly with the size of the private patch. All curves saturate at the minibatch-only baseline once the intersection probability becomes 1.

## 5.2 Influence of Crop and Patch Size on Privacy

We now analyze how the privacy parameter $\varepsilon$ varies with crop size and private patch size for three different noise levels. These factors determine the inclusion probability of the private region, which directly affects the effective sampling rate $\gamma_{\text{eff}} = \gamma_{\text{wo}} \cdot \gamma_{\text{crop}}$. We apply Theorem 2 to compute $\varepsilon$ from $\gamma_{\text{eff}}$ using PLD accounting. In the first experiment, we vary the crop size with a fixed private patch size of $10 \times 10$. In the second, we fix the crop size at $450 \times 450$ and vary the private patch size from 1 to 120 pixels. We set $\delta = 1/\texttt{epoch\_size}$ and assume $\texttt{epoch\_size} = 10^5$. In this section, we report results with Gaussian noise multipliers $\sigma \in \{4.0, 4.5, 5.0\}$, which keep $\varepsilon$ in a practically relevant interval and yield clearly interpretable plots. In Appendix D.2, we repeat the experiments with a wider range of noise values. All findings remain consistent with the ones shown here.

Figure 3 shows that $\varepsilon$ increases rapidly with crop size. In standard training, crop size is a hyperparameter that affects model performance. Our evaluation further indicates that it also influences privacy: larger crops yield higher $\varepsilon$, which in turn requires stronger noise for a fixed privacy budget. This creates an additional link between crop size and model performance. Figure 4 shows an approximately linear increase in $\varepsilon$ as private patch size grows. An important point is that both curves saturate at a cutoff, reaching the minibatch subsampling baseline. This occurs when the intersection probability reaches 1, meaning the private patch is included in every crop and contributes to the gradient whenever the image is selected. In this regime, patch-level subsampling becomes equivalent to standard minibatch sampling.

These results show that the strength of patch-level amplification depends critically on the inclusion probability of the private region, determined jointly by crop size, which should be tuned, and the (fixed) size of the private patch in the dataset. Additional results on the effect of image padding are provided in Appendix D.3.

### 5.3 Privacy-Utility Tradeoffs with Patch-Level Sampling

We conclude our experiments by evaluating whether the amplification provided by patch-level sampling improves model utility under a fixed privacy budget. We consider two widely used segmentation architectures, DeepLabV3+ (Chen et al., 2018) and PSPNet (Zhao et al., 2017), each trained on two datasets: Cityscapes (Cordts et al., 2016) and A2D2 (Geyer et al., 2020). For each model-dataset combination, we compare two variants of DP-SGD: one using standard minibatch subsampling, and one using our proposed patch-level sampling on top. Importantly, both settings use *exactly the same training algorithm*; the only difference lies in how privacy is analyzed. This means our method can be "switched on" without altering hyperparameters or implementation, and it will, in principle, always yield at least as much utility, either by allowing more training iterations for the same privacy budget or by requiring less noise to meet a given budget and epoch number.

To make this comparison fair and controlled, we adopt a single fixed hyperparameter configuration across all experiments. This configuration was chosen based on preliminary tuning and then held constant throughout, ensuring consistency across model-dataset pairs. While this sacrifices peak performance in individual cases, it isolates the effect of the privacy analysis itself. Each experiment is repeated with four random seeds, and we report the mean and standard deviation, depicted as error bars in Figure 5. Full details of the training are given in Appendix C.4.

As shown in Figure 5, patch-level sampling consistently yields higher utility for a given privacy-level $\varepsilon$. On Cityscapes with DeepLabV3+, it improves mean Intersection over Union (mIoU) by an average of over 40% across the tested range of $\varepsilon$, with gains as large as 81% at $\varepsilon \approx 5$. For PSPNet on the same dataset, the advantage is even more pronounced, with an average improvement of over 110% and up to 330% at $\varepsilon \approx 5$. On A2D2, we present results for PSPNet, where patch-level sampling improves mIoU by an average of 18% and up to 23% at $\varepsilon \approx 5$. We observe some variability across random initializations for DP-SGD, particularly with DeepLabV3+ on Cityscapes, which results in partially overlapping error bars between methods; per-seed results are provided in Appendix C.5. Additionally, under our compute constraints, DeepLabV3+ did not consistently converge on A2D2, and we therefore defer a detailed discussion of this setting to Appendix C.6.

We also compare with Gaussian noise augmentation from Schuchardt et al. (2025) as an alternative amplification mechanism in Appendix C.7; it performs worse than both our method and standard minibatch subsampling due to the high input sensitivity.

Lastly, we evaluate on image classification to show that the amplification effect generalizes beyond segmentation; results are provided in Appendix C.8. Since our method improves privacy accounting rather than training, the benefit is task-agnostic whenever random cropping is part of the pipeline.

Together, these results confirm that patch-level amplification enables substantially more favorable privacy-utility tradeoffs, especially in the low-to-moderate privacy budget regime where standard DP-SGD suffers most from noise-induced degradation.

## 6 Limitations and Future Work

Our work introduces patch-level DP as a way to exploit spatial structure in vision data. While our results demonstrate clear theoretical and empirical advantages over standard minibatch sampling, several limitations remain.

First, our analysis relies on a binary treatment of patch intersection: a patch is either included or not, regardless of the fraction of overlap. This may be conservative when partial overlap truly leaks less information. However, this is a fundamental property of DP-SGD: since the gradient is an arbitrary function of the input, we cannot assume that partial visibility leaks less than full visibility. Tighter guarantees could potentially be achieved with additional knowledge about the model, such as Lipschitz continuity of gradients (Das et al., 2023; Béthune et al., 2024), but our focus is on providing a general tool that works for arbitrary models.

Second, we assume that the size of the private patch is fixed and known in advance. If the assumed patch size is *larger* than the true sensitive region, our guarantees remain valid but become more conservative,

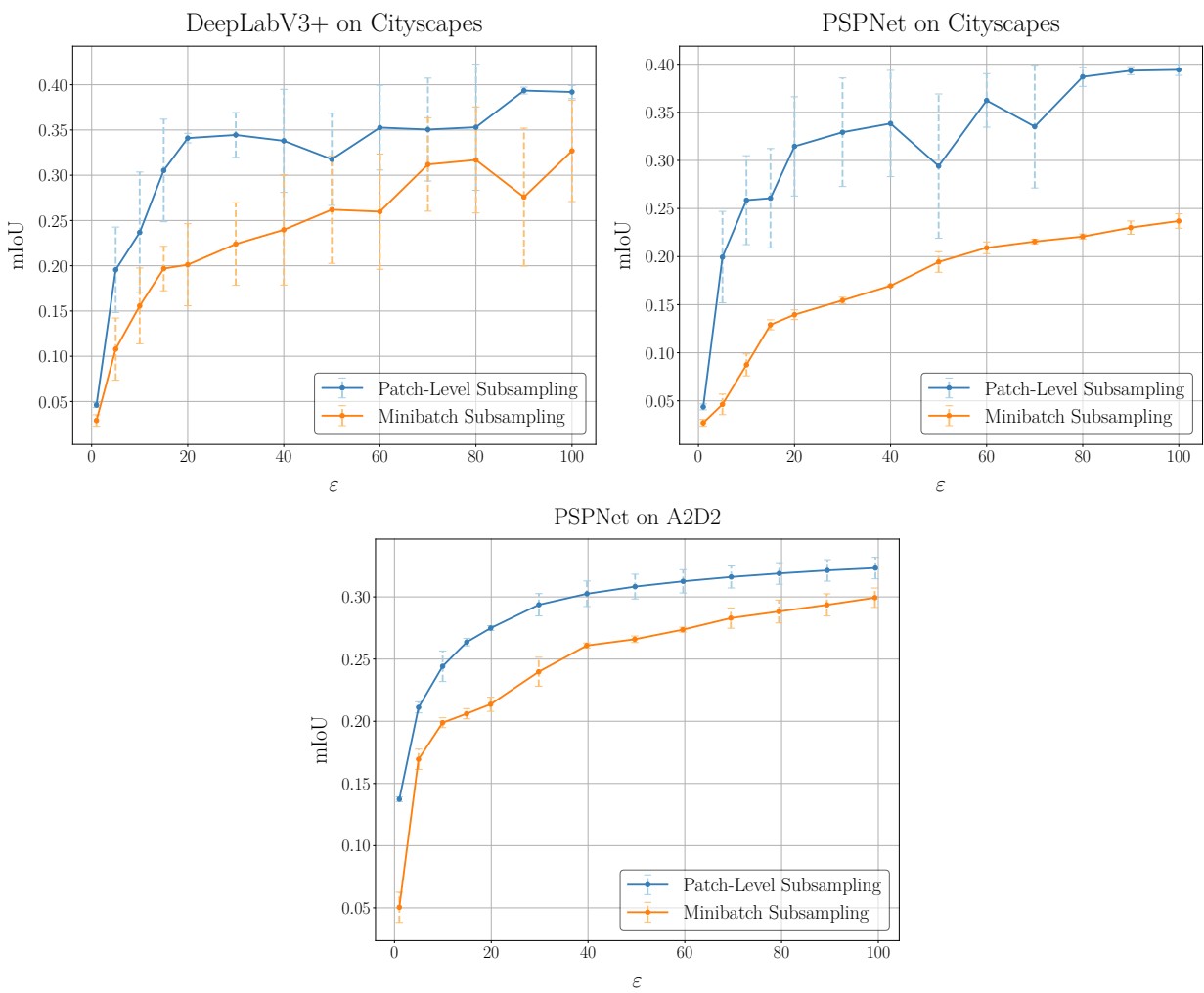

Figure 5: Model performance versus privacy-level $\varepsilon$ for DP-SGD with patch-level sampling (blue) and mini-batch subsampling (orange). Results are averaged over four seeds; error bars show standard deviation. For Cityscapes, we use $\delta = 1/2975$, and for A2D2, $\delta = 1/18557$, following $\delta = 1/$`epoch_size`. For both datasets a private patch size of $10 \times 10$ is assumed. DG-SGD with patch-level privacy overperforms significantly, given the exact same setup.

potentially sacrificing some utility. In the limit, if we assume the entire image is sensitive, we recover minibatch subsampling and standard record-level DP-SGD guarantees. If the assumed patch size is *smaller* than the true sensitive region, the stated guarantee can be too optimistic. However, this is general limitation in differential privacy, analogous to e.g. assuming record-level privacy when user-level privacy is required. In practice, practitioners should choose a conservative upper bound on the sensitive region size based on the amount of domain knowledge they have of their application. Crucially, our analysis is decoupled from the mechanism: given any combination of crop size and private patch size, we can compute the corresponding privacy guarantee without modifying training. Developing adaptive strategies for patch size remains an open question. One potential direction is detection-based patch modeling using domain-specific detectors (e.g., face or license plate detection), though such detectors must themselves be DP-trained or use public data to preserve guarantees. Importantly, when in doubt, practitioners should conservatively assume larger patch sizes, or in the limit, default to standard record-level DP-SGD, which is always a valid choice.

Finally, our method is applicable when sensitive content is spatially localized and random cropping is a natural part of the training pipeline. This is typically the case for high-resolution images where cropping

improves generalization. Conversely, our method does not apply when cropping is impractical (e.g., on low-resolution datasets like MNIST (LeCun et al., 2002) or CIFAR-10 (Krizhevsky et al., 2009) where cropping destroys semantic content; we report the emprical results on MNIST in Appendix C.8), or when crop size approaches image size, eliminating any exclusion probability. Many important real-world applications satisfy our requirements, including autonomous driving, medical imaging, surveillance, and satellite imagery.

Overall, we view these challenges not as drawbacks but as opportunities. Our framework opens the door to more sophisticated treatments of spatial privacy, such as overlap-aware accounting, adaptive private regions, and domain-specific extensions beyond vision. Exploring these directions could further strengthen the connection between differential privacy and the structured nature of real-world data.

## 7    Conclusion

In this work, we extended the traditional analysis of DP-SGD by considering *patch-level differential privacy* to better reflect the spatial structure of vision data. We formalized a patch-level neighboring relation and analyzed how random cropping can serve as a privacy amplification mechanism when private information is spatially localized. Our theoretical results show that the effective sampling rate decomposes into independent image- and patch-level components, generalizing classical amplification bounds. Empirically, we demonstrated that patch-level sampling consistently improves the privacy-utility trade-off across multiple architectures, datasets, and tasks. The approach is fully compatible with existing DP-SGD implementations and introduces no additional computational overhead. More broadly, our findings demonstrate that aligning privacy mechanisms with domain-specific structure and leveraging stochastic elements already present in training pipelines is a powerful approach, enabling the design of learning systems that are simultaneously more private and more accurate.

**Acknowledgements**

This paper was supported by the DAAD programme Konrad Zuse Schools of Excellence in Artificial Intelligence, sponsored by the Federal Ministry of Research, Technology and Space. It was further funded by the German Research Foundation (grant GU 1409/4-1).

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

## A Comparison of Patch-Level and Record-Level Privacy

The choice of neighboring relation is a fundamental modeling decision in differential privacy. In the vision domain, standard record-level privacy (Dwork et al., 2006; Dwork, 2006) assumes each image is a record, and changing a single attribute of a record is treated identically to changing all attributes. This is because it is assumed that the gradient is an arbitrary function of the input, and the only information available for privacy accounting is the clipping norm. We cannot assume that small input changes yield small gradient changes. So changing a single pixel in an image constitutes adjacency.

However, record-level privacy can be overly pessimistic when sensitive content is spatially localized. This is analogous to the distinction between user-level (Levy et al., 2021) and record-level privacy. In user-level privacy, neighboring datasets differ in all records belonging to a single user, whereas in record-level privacy, they differ in only a single record. User-level provides stronger protection but is more conservative; record-level is finer-grained and yields tighter guarantees when each user contributes exactly one record.

Our method extends this hierarchy by exploiting the spatial structure of vision data, introducing patch-level privacy as a finer-grained notion when sensitive content is localized. Patch-level DP is appropriate when privacy-sensitive information is confined to spatial regions whose size can be estimated or (conservatively) bounded. Typical examples include faces in photographs, license plates in traffic scenes, or lesions in medical imaging.

Thus, our work can be interpreted as weakening the worst-case assumption of record-level DP: instead of assuming the entire image may change, we assume only a bounded region may change (patch-level). Analogous to record-level privacy, a single pixel change in the private patch constitutes adjacency. When the domain structure supports this assumption, tighter guarantees follow.

In contrast, when the location of sensitive content cannot be bounded, record-level privacy should be used. Importantly, our method gracefully handles this case: as the private patch size approaches the full image size, patch-level privacy naturally converges to record-level privacy with minibatch subsampling (see Figure 4).

Finally, it is important to consider that coarser-grained relations always imply finer-grained ones: standard record-level privacy guarantees automatically provide patch-level privacy, since any patch substitution is a special case of record substitution. Standard record-level privacy is always a valid and conservative choice when no such assumptions can be safely made.

## B Proofs of patch-level privacy

### B.1 Proof of Lemma 3

Let the original image have width $W_I$ and height $H_I$, and let the crop have width $W_C$ and height $H_C$. Padding is applied symmetrically with $\text{pad}_x$ pixels horizontally and $\text{pad}_y$ pixels vertically. After padding, the number of valid horizontal and vertical crop origins are $W_{\text{tot}} = W_I + 2\text{pad}_x - W_C + 1$, $\quad H_{\text{tot}} = H_I + 2\text{pad}_y - H_C + 1$, as images form a discrete space by virtue of their pixel structure. The total number of possible positions of the crop $|\Omega|$ can be written as the total area $A_{\text{tot}} = W_{\text{tot}} \cdot H_{\text{tot}}$.

Let $R \subseteq \mathbb{N}_0^2$ be a fixed rectangular patch in the *original* image with bottom-left coordinate $(R_x, R_y)$ and size $H_R \times W_R$. After padding, its bottom-left coordinate becomes $R'_x = R_x + \text{pad}_x$, $R'_y = R_y + \text{pad}_y$.

A crop region $C_{u,v}$ of size $H_C \times W_C$ intersects $R$ with a minimum of a single pixel if and only if its origin $(u, v)$ satisfies $u \in [X_{\min}, X_{\max}]$, $v \in [Y_{\min}, Y_{\max}]$, where

$$X_{\min} = \max\left(0, R'_x - W_C + 1\right), \qquad X_{\max} = \min\left(W_I + 2\text{pad}_x - W_C, R'_x + W_R - 1\right),$$
$$Y_{\min} = \max\left(0, R'_y - H_C + 1\right), \qquad Y_{\max} = \min\left(H_I + 2\text{pad}_y - H_C, R'_y + H_R - 1\right),$$

as the image boundaries constrain the valid horizontal and vertical placements of the crop origin relative to the padded patch.

The number of crop origins that intersect $R$, meaning $|\Omega_R|$, can be written as the favorable area $A_{\text{fav}} = (X_{\max} - X_{\min} + 1) \cdot (Y_{\max} - Y_{\min} + 1)$. As we define the random cropping as a uniform distribution (Definition

5), all crop origins are equally likely. Thus, the patch-level inclusion probability is

$$\gamma'_{\text{crop}}(R) = \frac{A_{\text{fav}}}{A_{\text{tot}}} = \frac{(X_{\max} - X_{\min} + 1)(Y_{\max} - Y_{\min} + 1)}{(W_I + 2\text{pad}_x - W_C + 1)(H_I + 2\text{pad}_y - H_C + 1)}.$$

## B.2    Proof of Theorem 1

Given $\gamma'_{\text{crop}}(R)$ from Lemma 1, we want to maximize the patch-level inclusion probability. The denominator is constant, so it suffices to maximize $A_{\text{fav}}(R) := (X_{\max} - X_{\min} + 1)(Y_{\max} - Y_{\min} + 1)$. As defined previously, the horizontal factor $X_{\max} - X_{\min} + 1$ equals $\min(W_I + 2\,\text{pad}_x - W_C,\ R'_x + W_R - 1) - \max(0,\ R'_x - W_C + 1) + 1$, a function of $R'_x$ only. Consider four cases:

**Left-bound only.** If $R'_x - W_C + 1 < 0$ and $R'_x + W_R - 1 \leq W_I + 2\,\text{pad}_x - W_C$, then

$$X_{\max} - X_{\min} + 1 = (R'_x + W_R - 1) - 0 + 1 = R'_x + W_R,$$

which is strictly increasing in $R'_x$ for $R'_x \leq W_I + 2\,\text{pad} - W_C - W_R + 1$. Moving the patch to the right linearly increases the horizontal contribution.

**Right-bound only.** If $R'_x - W_C + 1 \geq 0$ and $R'_x + W_R - 1 > W_I + 2\,\text{pad}_x - W_C$, then

$$X_{\max} - X_{\min} + 1 = (W_I + 2\,\text{pad}_x - W_C) - (R'_x - W_C + 1) + 1 = W_I + 2\,\text{pad}_x - R'_x,$$

which is strictly decreasing in $R'_x$ for $R'_x \geq W_C - 1$. Moving the patch to the left linearly increases the horizontal contribution.

**Fully contained.** If $0 \leq R'_x - W_C + 1$ and $R'_x + W_R - 1 \leq W_I + 2\,\text{pad}_x - W_C$, then

$$X_{\max} - X_{\min} + 1 = (R'_x + W_R - 1) - (R'_x - W_C + 1) + 1 = W_R + W_C - 1,$$

which is constant in $R'_x$ for $W_C - 1 \leq R'_x \leq W_I + 2\,\text{pad}_x - W_C - W_R + 1$. Any horizontal shift within this range leaves the horizontal contribution unchanged at its maximum value.

**Both bounds active.** If $R'_x - W_C + 1 < 0$ and $R'_x + W_R - 1 > W_I + 2\,\text{pad}_x - W_C$, which can occur only if $W_I + 2\,\text{pad}_x < 2W_C + W_R - 2$, then

$$X_{\max} - X_{\min} + 1 = (W_I + 2\,\text{pad}_x - W_C) - 0 + 1 = W_I + 2\,\text{pad}_x - W_C + 1,$$

which is constant in $R'_x$ for all positions satisfying the inequalities above. In this regime, every admissible horizontal placement achieves the largest possible horizontal value, and the maximizer is determined solely by the vertical factor.

From the four regimes above, we conclude that only (1) in the left-bound regime, shifting $R$ right increases the horizontal factor until the left bound is inactive. (2) In the right-bound only regime, shifting $R$ left increases the horizontal factor until the right bound is inactive. (3) Once both bounds are inactive (fully contained), the horizontal factor is constant at its maximum possible value within unclipped regimes. (4) In the case that both bounds are active, the horizontal factor attains its absolute maximum and is independent of $R'_x$. In all cases, the horizontal factor is maximized when the patch is positioned as far from both borders as possible. When the width allows an unclipped regime, placing the patch in the middle ensures it lies in that regime; the same holds in the special case where both bounds are active. In edge cases where no constant regime exists, every admissible position already achieves the maximum.

By symmetry, the vertical factor is optimized by the same reasoning, placing $R$ centrally along the vertical axis. Combining both axes yields the centrally placed patch, whose coordinates in the original image are

$$R_{\text{center}} = \left( \left\lfloor \frac{H_I - H_R}{2} \right\rfloor, \ \left\lfloor \frac{W_I - W_R}{2} \right\rfloor \right),$$

which maximizes $\gamma'_{\text{crop}}(R)$.

## B.3 Proof of Theorem 2

In the following, we derive the dominating pair for patch-level subsampled mechanism $\mathcal{M} \circ S^{\text{crop}} \circ S_m^{\text{wo}}$, where $M : \mathcal{X} \to \mathbb{R}^D$ is an arbitrary mechanism, i.e., $M(x)$ is a distribution over co-domain $\mathbb{R}^D$ given a (subsampled) dataset $x \in \mathcal{X}$.

To characterize the output distribution of the patch-level subsampled mechanism, we first observe that cropping and dataset-level subsampling commute, meaning we can first sample a random crop for each image and then select a subset of images. Thus, the distribution for a specific batch can be fully characterized by dataset $x = \{x_1, \ldots, x_n\}$, the subset of selected indices $\mathbb{J} \subseteq [n]$, and the sequence of selected crop origins $o \in \Omega^n$.[2] With a slight abuse of notation, we denote this distribution as $M(x, J, o)$. Thus, for a batch size $m$, the overall patch-level subsampled distribution is

$$(\mathcal{M} \circ S^{\text{crop}} \circ S_m^{\text{wo}})(x) = \sum_{\mathbb{J} \subseteq [n] | |\mathbb{J}| = m} \binom{n}{m}^{-1} \sum_{o \in \Omega^n} \left(\frac{1}{|\Omega|}\right)^n M(x, \mathbb{J}, o). \tag{4}$$

In the following, we consider w.l.o.g. a patch-level neighboring dataset $x' \simeq_{\Delta,p}$ which differ within some worst-case patch $R$ within the $n$th image, i.e., $x' = \{x_1, x_2, \ldots, x_n'\}$. Under this assumption, we can separate out parts of the subsampling distribution that are identical between $x$ and $x'$, and those are that dissimilar: We observe that the generative process defined by 4 is equivalent to

1. sampling crop origins $o_{-n} \in \Omega^{n-1}$ for the first $n-1$ images
2. sampling uniformly at random a subset $\mathbb{J}_{-n}$ of size $m-1 \subseteq [n-1]$ from the first $n-1$ images and an arbitrary

and then either

(a) with probability $1 - \gamma_{\text{wo}}$, sampling uniformly at random an index $a \in [n-1] \setminus \mathbb{J}$ to include in the batch and selecting an arbitrary $n$th crop origin $o_n$
(b) with probability $\gamma_{\text{wo}}(1 - \gamma'_{\text{crop}}(R))$, including the $n$th image in the batch and sampling uniformly at random a crop origin $o_{n, \neg R} \in \Omega_{\neg R}$ that does not intersect with $R$
(c) with probability $\gamma_{\text{wo}} \gamma'_{\text{crop}}(R)$, including the $n$th image in the batch and sampling uniformly at random a crop origin $o_{n,R} \in \Omega_R$ that intersects with $R$

Thus, we can restate Eq. 4 as

$$\sum_{\mathbb{J}_{-n} \subseteq [n-1] | |\mathbb{J}| = m-1} \binom{n-1}{m-1}^{-1} \sum_{o_{-n} \in \Omega^{n-1}} \left(\frac{1}{|\Omega|}\right)^{n-1} \sum_{a \in [n-1] \setminus \mathbb{J}_{n-1}} \frac{1}{n-m} \sum_{o_{n,\neg R} \in \Omega_{\neg R}} \sum_{o_{n,R} \in \Omega_R} \frac{1}{|\Omega_{\neg R}| \cdot |\Omega_R|} \cdot \Big($$
$$(1 - \gamma_{\text{wo}}) \cdot M(x, \mathbb{J}_{-n} \cup \{a\}, (o_{-n}, o_n))$$
$$+ \gamma_{\text{wo}} \cdot (1 - \gamma'_{\text{crop}}(R)) \cdot M(x, \mathbb{J}_{-n} \cup \{n\}, (o_{-n}, o_{n,\neg R}))$$
$$+ \gamma_{\text{wo}} \cdot \gamma'_{\text{crop}}(R) \cdot M(x, \mathbb{J}_{-n} \cup \{n\}, (o_{-n}, o_{n,R}))\Big)$$

For neighboring dataset $x' \simeq_{\Delta,p}$, we can restate $(\mathcal{M} \circ S^{\text{crop}} \circ S_m^{\text{wo}})(x')$ in an analogous manner by replacing every occurrence of $x$ with $x'$. Since both mixture distributions share identical mixture weights, the following result follows immediately from the joint convexity (Balle et al., 2020) and thus joint quasi-convexity of hockey-stick divergence $H_\alpha$:

**Lemma 2.** *The patch-level subsampled mechanism* $(\mathcal{M} \circ S^{\text{crop}} \circ S_m^{\text{wo}})$ *is dominated by the three-component mixture mechanism*

$$\tilde{M}(x) = (1 - \gamma_{wo})M_{\neg n}(x) + \gamma_{wo} \cdot (1 - \gamma'_{\text{crop}}(R))M_{n,\neg R}(x) + \gamma_{wo} \cdot \gamma'_{\text{crop}}(R)M_{n,R}(x) \tag{5}$$

---

[2]The selected origins are chosen to be a sequence to define a correspondence between crops and images.

*with mixture components*

$$M_{\neg n}(x) = M(x, \mathbb{J}_{-n} \cup \{a\}, (o_{-n}, o_n))$$
$$M_{n, \neg R}(x) = M(x, \mathbb{J}_{-n} \cup \{n\}, (o_{-n}, o_{n, \neg R}))$$
$$M_{n, R}(x) = M(x, \mathbb{J}_{-n} \cup \{n\}, (o_{-n}, o_{n, R}))$$

*for some index $a \in [n-1]$, and some crop origns $o_n \in \Omega$, $o_{n, \neg R} \in \Omega_{\neg R}$, and $o_{n, \neg R} \in \Omega_R$.*

More simply stated, $M_{\neg n}$ refers to the the mechanism for a specific subset of indices in which the $n$th image is not included. Next, $M_{n, \neg R}$ is the mechanism for a specific batch in which the $n$th image is included but the private region $R$ is not in the crop. Finally, $M_{n, R}$ is the mechanism for a subset of indices in which the $n$th image is included and the private region $R$ is in the crop. Overall, Lemma 2 lets us reduce the analysis of our high-dimensional mixture mechanism to a mixture mechanism with just three components.

In the following, we will derive privacy guarantees for this simplified mechanism in terms of dominating pairs of the underlying mechanism $M : \mathcal{X} \to \mathbb{R}^D$, beginning with $\alpha \geq 1$ and then proceeding to $0 \leq \alpha < 1$.

**Case 1: $\boldsymbol{\alpha \geq 1}$** For our proof, we will expand upon the following "advanced joint convexity" property (Balle et al., 2018), which allows for a simplified analysis of two-component mixtures where the first component is identical.

**Lemma 3** (Advanced joint convexity). *Let $P = (1-\eta)P_1 + \eta P_2$ and $Q = (1-\eta)Q_1 + Q_2$ be two mixture distributions with $P_1 = Q_1$ and some $\eta \in [0, 1]$. Given $\alpha \geq 1$, let $\alpha' = (\alpha - 1)/\eta + 1$ and $\beta = \alpha / \alpha'$. Then, the following holds:*
$$H_\alpha(P||Q) = \eta H_{\alpha'}(P_2||(1-\beta)P_1 + \beta Q_2).$$

Specifically, the advanced joint convexity property can also be used to analyze three-component mixtures, where the first three components are identical, which is the case when applying $\tilde{M}$ from Eq. 5 to $x$ and $x'$:

**Lemma 4.** *Let $P = \eta_1 P_1 + \eta_2 P_2 + \eta_3 P_3$ and $Q = \eta_1 Q_1 + \eta_2 Q_2 + \eta_3 Q_3$ be two mixture distributions with $P_1 = Q_1$, $P_2 = Q_2$ and some $\boldsymbol{\eta} \in [0, 1]^3$ with $||\boldsymbol{\eta}||_1 = 1$. Given $\alpha \geq 1$, let $\alpha' = (\alpha - 1)/\eta_3 + 1$ and $\beta = \alpha/\alpha'$. Then, the following holds:*

$$H_\alpha(P||Q) \leq \eta_3 \max\{H_{\alpha'}(P_3||P_1), H_{\alpha'}(P_3||P_2), H_{\alpha'}(P_3||Q_3)\}$$

*Proof.* The distributions $P$ and $Q$ can be trivially reformulated as

$$P = (\eta_1 + \eta_2)\left(\frac{\eta_1}{\eta_1 + \eta_2}P_1 + \frac{\eta_2}{\eta_1 + \eta_2}P_2\right) + \eta_3 P_3,$$
$$Q = (\eta_1 + \eta_2)\left(\frac{\eta_1}{\eta_1 + \eta_2}Q_1 + \frac{\eta_2}{\eta_1 + \eta_2}Q_2\right) + \eta_3 Q_3.$$

Using $P_1 = Q_1$ and $P_2 = Q_2$ and applying Lemma 3 then shows

$$H_\alpha(P||Q) = \eta_3 H_{\alpha'}\left(P_3||(1-\beta)\cdot\left(\frac{\eta_1}{\eta_1 + \eta_2}P_1 + \frac{\eta_2}{\eta_1 + \eta_2}P_2\right) + \beta Q_3.\right).$$

The result then follows from quasi-convexity of the hockey stick divergence in the space of measures. □

Combining 2 and 4, we can complete our analysis for $\alpha \geq 1$:

**Lemma 5.** *Let $\mathcal{M}$ be a mechanism that is dominated by $P, Q$ under the substitution relation $\simeq_\Delta$. Let $S_m^{\mathrm{wo}}$ denote without-replacement subsampling of minibatch size $m$ and $S^{\mathrm{crop}}$ the random cropping mechanism we defined in Definition 5. Then, under the patch-level substitution relation $\simeq_{\Delta, p}$, the composed mechanism $\mathcal{M} \circ S^{\mathrm{crop}} \circ S_m^{\mathrm{wo}}$ satisfies*

$$\delta_{\mathcal{M} \circ S^{\mathrm{crop}} \circ S_m^{\mathrm{wo}}}(\alpha) \leq H_\alpha\left((1-\gamma_{\mathrm{eff}})Q + \gamma_{\mathrm{eff}}P \,\|\, Q\right) \tag{6}$$

*for any $\alpha \geq 1$.*

*Proof.* Consider an arbitrary pair of datasets $x \simeq_{\Delta,p} x'$ that differ in an arbitrary patch $R$. From 2 and 4, it immediately follows that

$$H_\alpha((\mathcal{M} \circ S^{\mathrm{crop}} \circ S_m^{\mathrm{wo}})(x)||(\mathcal{M} \circ S^{\mathrm{crop}} \circ S_m^{\mathrm{wo}})(x'))$$
$$\leq \gamma_{\mathrm{wo}} \cdot \gamma'_{\mathrm{crop}}(R) \cdot \max \left\{ H_{\alpha'}(M_{n,R}(x')||M_{\neg n}(x), H_{\alpha'}(M_{n,R}(x')||M_{n,\neg R}(x), H_{\alpha'}(M_{n,R}(x')||M_{n,R}(x) \right\}$$

with $\alpha' = (\alpha - 1) / (\gamma_{\mathrm{wo}} \cdot \gamma'_{\mathrm{crop}}(R)) + 1$

It is easy to see that in all three cases, the batches operated on by both mechanisms only differ by a single substitution:

1. $M_{\neg n}(x)$ is equivalent to $M_{n,R}(x')$ after removing the cropped $x'_n$ from the batch and replacing it with some $x_a$.
2. $M_{n,\neg R}(x)$ is equivalent to $M_{n,R}(x')$ after replacing the cropped $x'_n$ with a differently cropped $x_n$.
3. $M_{n,R}(x)$ is equivalent to $M_{n,R}(x')$ after replacing the cropped $x'_n$ with an identically cropped $x_n$.

It thus follows that

$$H_\alpha((\mathcal{M} \circ S^{\mathrm{crop}} \circ S_m^{\mathrm{wo}})(x)||(\mathcal{M} \circ S^{\mathrm{crop}} \circ S_m^{\mathrm{wo}})(x')) \leq \gamma_{\mathrm{wo}} \cdot \gamma'_{\mathrm{crop}}(R) \cdot H_{\alpha'}(P||Q).$$

Next, we observe that this term is maximized by patch that maximizes the intersection probability, i.e., $\arg\max_R \gamma'_{\mathrm{crop}}(R)$. This is because $\alpha'$ is monotonically decreasing in and $H_{\alpha'}(P||Q)$ is in itself monotonically decreasing in $\alpha'$. After substituting $\gamma_{\mathrm{sub}} \cdot \max_R \gamma'_{\mathrm{crop}}(R)$ with $\gamma_{\mathrm{eff}}$, the final step is to apply advanced joint convexity in reverse order with $\beta = \alpha / \alpha'$:

$$\gamma_{\mathrm{eff}} \cdot H_{\alpha'}(P||Q) = \gamma_{\mathrm{eff}} \cdot H_{\alpha'}(P||(1-\beta)Q + \beta Q) = H_{\alpha'}((1-\gamma_{\mathrm{eff}})Q + \gamma_{\mathrm{eff}}P||(1-\gamma)Q + \gamma Q).$$

Substituting $(1-\gamma)Q + \gamma Q = Q$ concludes our proof. $\square$

**Case 2: $\alpha \geq 1$** The proof for this case largely follows the final steps of the proof of Proposition 30 in (Zhu et al., 2022). We will use the following Lemma, which corresponds to Lemma 31 from (Zhu et al., 2022):

**Lemma 6.** *Let $M$ be a mechanism and $\simeq$ be a symmetric neighboring relation. Then*

1. *If $(P,Q)$ is a dominating pair of $M$, then $(Q,P)$ is also a dominating pair of $M$*

2. *The following two statements are equivalent:*

   (a) $\sup_{x \simeq x'} H_\alpha(M(x)||M(x')) \leq H_\alpha(P||Q)$ *for all $\alpha \geq 1$.*
   (b) $\sup_{x \simeq x'} H_\alpha(M(x)||M(x')) \leq H_\alpha(Q||P)$ *for all $0 \leq \alpha < 1$.*

We can use the first part of Lemma 6 to conduct exactly the same proof as for 5, but interchange all occurrences of $P$ and $Q$. The following result then immediately follows from the second part of Lemma 6:

**Lemma 7.** *Let $\mathcal{M}$ be a mechanism that is dominated by $P, Q$ under the substitution relation $\simeq_\Delta$. Let $S_m^{\mathrm{wo}}$ denote without-replacement subsampling of minibatch size $m$ and $S^{\mathrm{crop}}$ the random cropping mechanism from Definition 5. Then, under the patch-level substitution relation $\simeq_{\Delta,p}$, the composed mechanism $\mathcal{M} \circ S^{\mathrm{crop}} \circ S_m^{\mathrm{wo}}$ satisfies*

$$\delta_{\mathcal{M} \circ S^{\mathrm{crop}} \circ S_m^{\mathrm{wo}}}(\alpha) \leq H_\alpha \left( P \,\|\, (1-\gamma_{\mathrm{eff}})P + \gamma_{\mathrm{eff}}Q \right) \tag{7}$$

*for any $0 \leq \alpha < 1$.*

Combined with Lemma 5, this concludes our proof of Theorem 2.

### B.4 Tightness of Theorem 2 for DP-SGD

In the following, we prove that Eq.3 holds with equality for noisy, clipped gradient updates for some worst-case model.

Consider an arbitrary differentiable model $f : \mathcal{I}_C \to \mathcal{Y}$ that maps a single cropped image to a single prediction from some co-domain $\mathcal{Y}$. Assume that the model has $D \in \mathbb{N}$ parameters. Let $g : \mathcal{I}_C \to \mathbb{R}^D$ be the function that maps a single cropped image to the parameter gradients under some loss function. Like in Abadi et al. (2016), we can then define our differentially private gradient mechanism $M : 2^{\mathcal{I}_C} \to$ as

$$M(Z) = \frac{1}{|Z|} \left( \sum_{z \in Z} \text{Clip}(g(z), \kappa) + \mathcal{N}(\mathbf{0}, \sigma^2 \mathbf{1} \kappa^2) \right),$$

where $\text{Clip}(g(z), \kappa)$ scales the gradient to a maximum norm of $\kappa \in \mathbb{R}_+$.

Mechanism $M$ is simply a Gaussian mechanism with $\ell_2$ sensitivity 2 under substitution relation $\simeq_\Delta$ (note that the covariance is scaled by $\kappa^2$). Thus, analogous to Proposition 4, it is dominated by univariate Gaussians $\mathcal{N}(0, \sigma)$ and $\mathcal{N}(2, \sigma)$. Due to translation invariance of hockey-stick divergences between Gaussians, it is also dominated by $P = \mathcal{N}(-1, \sigma)$ and $Q = \mathcal{N}(1, \sigma)$.

To show that it is tightly dominated by $P, Q$ under the standard substitution relation, and since we do not have any additional information about gradients $g$ or model $f$, consider a worst-case $g : \mathcal{I}_C \to \mathbb{R}^D$ with $g(z) = \begin{bmatrix} \kappa & 0 \cdots 0 \end{bmatrix}^T$ if the value 0 appears in the input cropped image $z$ and $g(z) = \begin{bmatrix} -\kappa & 0 \cdots 0 \end{bmatrix}^T$ otherwise. For this specific $g$ define the cropped batches $Z = \{z_1, z_2, \ldots, z_N\}$ and $Z' = \{z_1, z_2, \ldots, z'_N\}$ where $z_i$ has value $i$ in every pixel and $z'$ has value 0 in every pixel. Then,

$$\begin{aligned}
\sup_{Z \simeq_\Delta Z'} H_\epsilon(M(Z) \| M(Z')) \geq & H_\epsilon(\mathcal{N}(-n\kappa e_1, \kappa^2 \sigma^2 \mathbf{1}) \| \mathcal{N}(-(n-1)\kappa e_1 + \kappa e_1, \kappa^2 \sigma^2 \mathbf{1}) \\
= & H_\epsilon(\mathcal{N}(-\kappa, \kappa\sigma) \| \mathcal{N}(\kappa, \kappa\sigma)) \\
= & H_\epsilon(\mathcal{N}(-1, \sigma) \| \mathcal{N}(1, \sigma)).
\end{aligned} \tag{8}$$

This lower bound for the non-subsampled base mechanism coincides with the upper bound given by the privacy profile of dominating pair $P, Q$.

As per Theorem 2, applying minibatch subsampling and cropping to mechanism $M$ to construct $\mathcal{M} : 2^{\mathcal{I}} \to \mathbb{R}^D$ yields a privacy-profile upper-bounded by

$$\delta_{\mathcal{M} \circ S^{\text{crop}} \circ S_m^{\text{wo}}}(\alpha) \leq \begin{cases} H_\alpha\left((1 - \gamma_{\text{eff}})Q + \gamma_{\text{eff}}P \,\|\, Q\right) & \text{if } \alpha \geq 1, \\ H_\alpha\left(P \,\|\, (1 - \gamma_{\text{eff}})P + \gamma_{\text{eff}}Q\right) & \text{if } 0 < \alpha < 1, \end{cases} \tag{9}$$

In the following, we construct a worst-case dataset for each $\alpha \geq 1$ such that the previously constructed gradient function $g$ has a privacy profile that exactly matches this upper bound.

**Case 1: $0 \leq \alpha < 1$** Define the original dataset of images $X = \{x_1, x_2, \ldots, x_n\}$ where the $i$th image has value $i$ everywhere. Define the patch-level substituted dataset of images $X' \simeq_{\Delta,p}$ that is identical to $X$ everywhere, except for all pixels of $x_n$ in worst-case patch from Theorem 1 being replaced by 0. By construction, the gradient function $g$ is only ever applied to cropped images that do not contain 0 and thus only ever returns $g(z) = \begin{bmatrix} -\kappa & 0 \cdots 0 \end{bmatrix}^T$. When operating on $X'$, the gradient function is applied to a cropped image containing value 0 with probability $\gamma \cdot \gamma_{\text{crop}}$, in which case it returns $g(z) = \begin{bmatrix} \kappa & 0 \cdots 0 \end{bmatrix}^T$.

Thus, and by using the translation and scale invariance of hockey-stick divergences between Gaussians in the same manner as in Eq. 8,

$$\delta_{\mathcal{M} \circ S^{\text{crop}} \circ S_m^{\text{wo}}}(\alpha) \geq H_\epsilon(\mathcal{N}(-1, \sigma) \| (1 - \gamma \cdot \gamma_{\text{crop}} \cdot \mathcal{N}(-1, \sigma) + \gamma \cdot \gamma_{\text{crop}} \cdot \mathcal{N}(1, \sigma)),$$

which matches Eq.9

**Case 2: $1 \leq \alpha$** The proof is fully analogous, one just needs to exchange the constructed $X$ and $X'$.

## C Experimental Setup

In Section 5, we described the main experiments and their results. Here, we provide the detailed configurations and discussion to ensure reproducibility and to clarify the precise conditions under which our results were obtained.

### C.1 Datasets

We evaluate on `Cityscapes` (Cordts et al., 2016) and `A2D2` (Geyer et al., 2020). Cityscapes contains 5000 finely annotated street-scene images at $1024 \times 2048$ resolution with a 2975/500 train/val split; since the official test labels are unavailable, we report on the validation set. A2D2 consists of 33,441 driving-scene images at $1208 \times 1920$, annotated for 18 classes (grouped from 38 original classes to create a Cityscapes-like taxonomy), and we use a session-based split with 11 training sessions, 3 validation sessions, and 3 test sessions.

### C.2 Models

We use `DeepLabV3+` (Chen et al., 2018)[3] and `PSPNet` (Zhao et al., 2017). For compatibility with DP-SGD, we disabled batch normalization layers in both architectures, as batch normalization relies on batch-level statistics, which are incompatible with privacy guarantees under DP training. Backbones are initialized from ImageNet (Deng et al., 2009); segmentation heads are randomly initialized. We fine-tune the entire model end-to-end.

### C.3 Differential Privacy Setup

Differential privacy is implemented with Opacus' (Yousefpour et al., 2022) `DPOptimizer` and accounted with privacy loss distribution (PLD) composition (Dong et al., 2022; Sommer et al., 2019) using the Google `dp_accounting` library (Google Differential Privacy Team, 2024). Minibatches are sampled uniformly without replacement via `DPDataLoader`. Gradients are clipped at per-sample $\ell_2$ norm. $\delta$ is set to 1/`epoch_size`. We quantize PLDs using the "connect-the-dots" algorithm (Doroshenko et al., 2022).

### C.4 Per-Experiment Configurations

**Mechanism-Specific Privacy Profiles.** We compare three mechanisms: (1) standard DP-SGD with minibatch subsampling, (2) patch-level subsampling applied on top of minibatching, and (3) a noise-addition baseline. The default setup uses image size $1000 \times 1000$, padding 0, clip norm 2.0, batch size 100, epoch size 3000, and number of epochs 100. Crop and patch sizes, as well as noise multipliers, are varied depending on the specific experiment (see Appendix D.1). We report $\delta$ as a function of $\varepsilon$.

**Influence of Crop and Patch Size.** We fix the same defaults as above and additionally set $\delta = 1/$`epoch_size` and `epoch_size` $= 10^5$. We perform two sweeps. First, we vary the crop side length from 50 to 700 pixels while keeping the private patch size fixed at $10 \times 10$. Second, we fix the crop size at $450 \times 450$ and vary the patch size from 1 to 50. We report $\varepsilon$ in both setups.

**Privacy-Utility Tradeoffs.** Following prior work (Ponomareva et al., 2023; Sander et al., 2022; De et al., 2022; McKenna et al., 2025), which emphasizes the strong dependence of DP-SGD performance on hyperparameters, we first conducted a tuning stage on Cityscapes with DeepLabV3+. The goal was to maximize utility at a fixed privacy budget of $\varepsilon = 10$ with $\delta = 1/2975$ and private patch size $10 \times 10$. This yielded a configuration with crop size $505 \times 505$, zero padding, batch size 200 (accumulated from microbatches of size 2 via Opacus' `BatchMemoryManager`), clipping norm $C = 2.0$, learning rate 0.02, and training horizon of 100 epochs. We then fixed these hyperparameters across all subsequent experiments, including different

---

[3]DeepLabV3+ is implemented based on the repository `https://github.com/VainF/DeepLabV3Plus-Pytorch`.

models and datasets, to ensure comparability and to isolate the effect of the privacy analysis. For A2D2, the only modification was reducing the training horizon to 25 epochs due to computational limits, with $\delta = 1/18{,}557$. For experiments spanning different privacy budgets, we kept the training horizon constant (100 epochs on Cityscapes, 25 on A2D2) and used `dp_accounting` (Google Differential Privacy Team, 2024) library to compute the Gaussian noise multiplier $\sigma$ corresponding to each target $\varepsilon$ (ranging from 5 to 100). This guarantees that results are reported under consistent training conditions, with $\sigma$ adapted to match the specified privacy-level. All experiments were repeated with four random seeds; we report the mean and standard deviation of the mean Intersection-over-Union (mIoU). We managed these experiments with `seml` (Zügner et al., 2023).

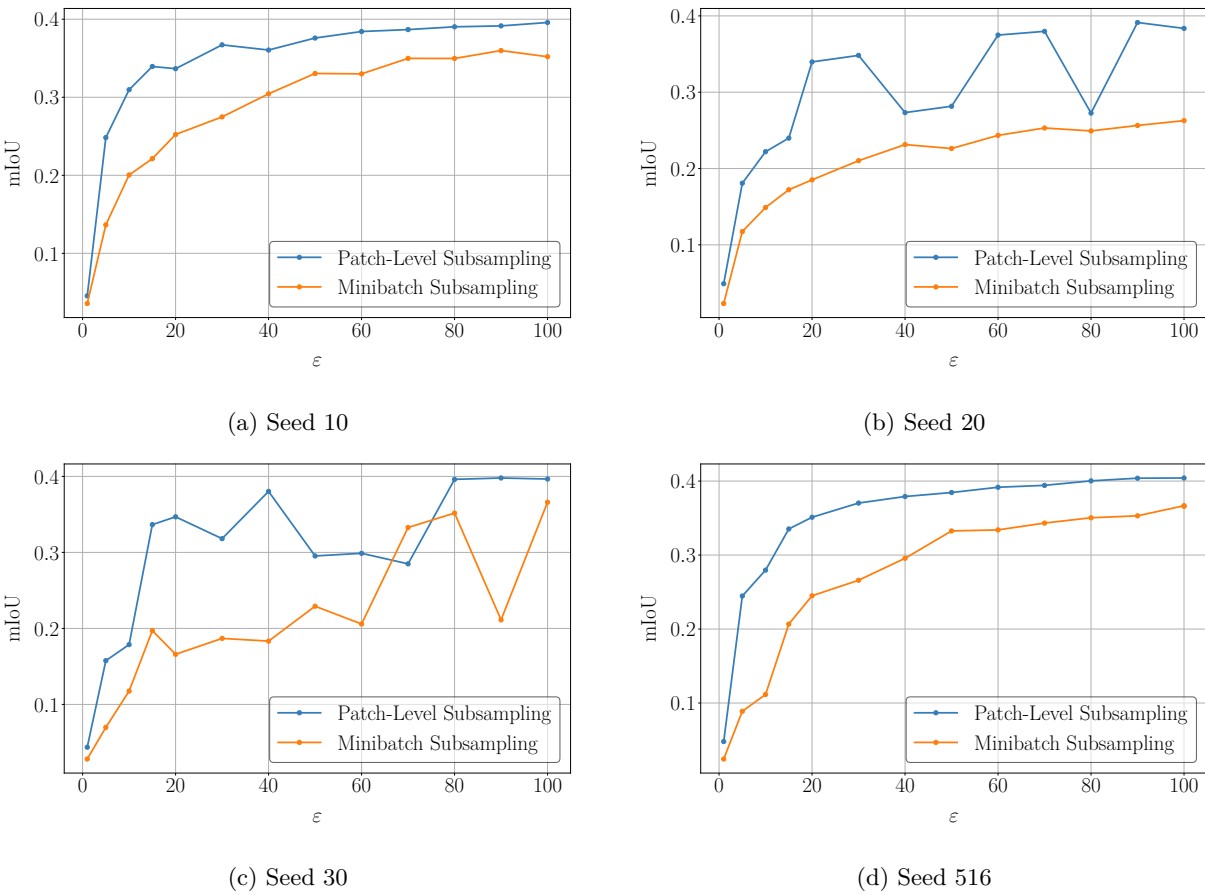

Figure 6: Per-seed privacy-utility tradeoffs of DeepLabV3+ on Cityscapes. We assume a private patch size of $10 \times 10$, and set $\delta = 1/2975$.

## C.5 Per-Seed Results for Privacy-Utility Tradeoffs

To further examine the high variability in the results of Figure 5, we report per-seed results for model–dataset combinations in Figures 6–8. These plots explicitly illustrate the variability across random initializations that is characteristic of DP-SGD training, as we mentioned in Appendix C.4. On Cityscapes with DeepLabV3+, this variability explains the overlapping error bars noted in the main text. In one seed, for $\varepsilon = 70$, the baseline briefly exceeds the patch-level sampling variant at a single privacy level, while the reverse holds across all other values and all remaining seeds. This behavior is typical as the performance of DP-SGD is highly parameter dependent, and the fixed hyperparameter configuration is optimized for high peak performance

rather than reduced variance. Across all seeds, patch-level sampling consistently achieves higher mean utility, which is what is reflected in the aggregated results shown in Figure 5.

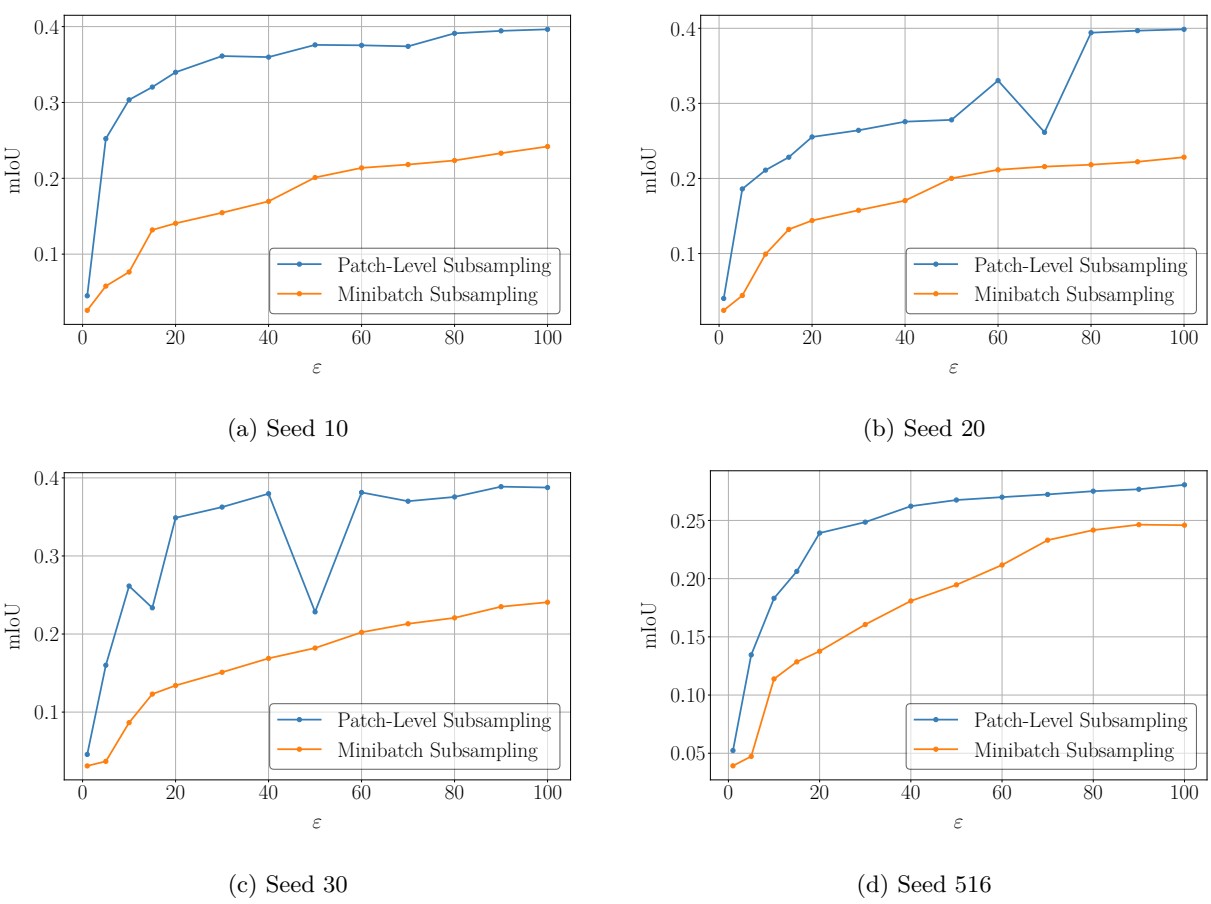

Figure 7: Per-seed privacy-utility tradeoffs of PSPNet on Cityscapes. We assume a private patch size of $10 \times 10$, and set $\delta = 1/2975$.

## C.6 Convergence for A2D2 with DeepLabV3+

We attempted training DeepLabV3+ on A2D2 under the fixed hyperparameter configuration described above. Unlike the other model-dataset combinations, however, this configuration did not consistently converge. Out of four random seeds, three produced stable training and yielded results consistent with our main conclusions, while one failed to make progress (see Figure 9).

Importantly, the same instability was also observed for the baseline method with standard minibatch subsampling. Thus, while we cannot make conclusive comparisons between the two sampling strategies in this setting, the divergence is not inherent to patch-level subsampling. Instead, we attribute it to a combination of (1) the reduced training horizon of 25 epochs imposed by dataset size and computational constraints, and (2) the lack of dataset-specific hyperparameter tuning. Given the sensitivity of DP-SGD training to hyperparameters (Ponomareva et al., 2023; Sander et al., 2022; De et al., 2022; McKenna et al., 2025), it is likely that adjustments (e.g., learning rate schedules or longer training) would resolve this issue, but such tuning was not feasible under our compute budget.

Figure 10 shows the per-seed results. While one run diverges, the remaining three clearly demonstrate the same utility gains from patch-level subsampling that we observe in the other settings. This suggests that the lack of convergence is a practical artifact of limited resources rather than a limitation of the method itself.

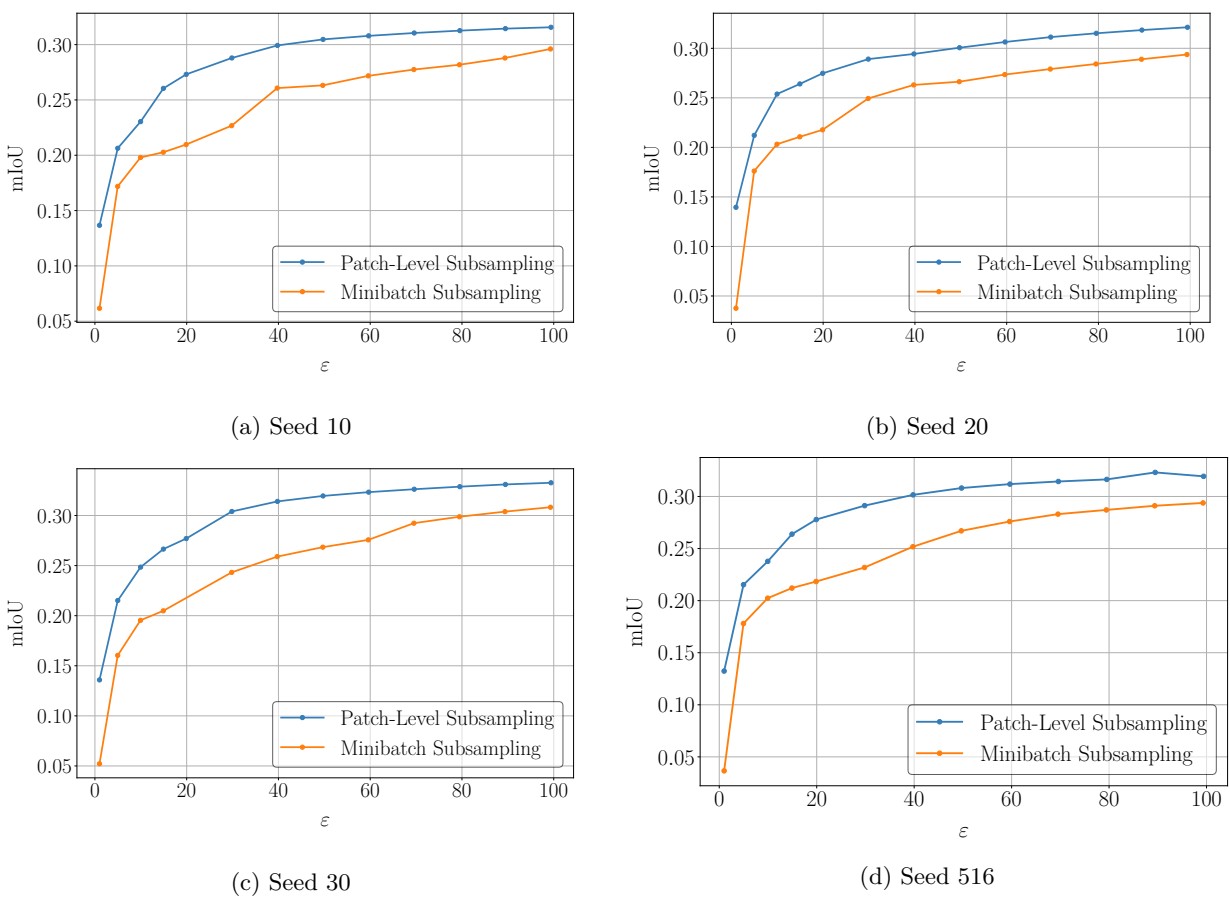

(a) Seed 10

(b) Seed 20

(c) Seed 30

(d) Seed 516

Figure 8: Per-seed privacy-utility tradeoffs of PSPNet on A2D2. We assume a private patch size of $10 \times 10$, and set $\delta = 1/18557$.

## C.7 Comparison with Gaussian Noise Augmentation

In the following, we compare our patch-level privacy amplification with Gaussian noise augmentation as an alternative amplification mechanism, following the analysis of Schuchardt et al. (2025). In this setup, Gaussian noise is added to images during training, and the resulting amplification is accounted for via TVD-based composition with minibatch subsampling. We calibrate the noise level to match the same target $\varepsilon$ values as our other experiments. Figures 11 and 12 shows results for DeepLabV3+ and PSPNet on Cityscapes and A2D2. Gaussian noise augmentation performs worse than both our method and standard minibatch subsampling. This is because the $\ell_2$ sensitivity of the input is large, so achieving meaningful amplification via TVD would require impractically high noise levels. When we instead calibrate the noise to match our target $\varepsilon$ values, the resulting amplification on top of minibatch subsampling is negligible, while the added noise still degrades model performance. This makes the model perform even worse than the minibatch subsampling baseline. In contrast, our patch-level analysis leverages the inherent randomness of cropping, which is already part of standard training pipelines and does not introduce additional perturbations to the input.

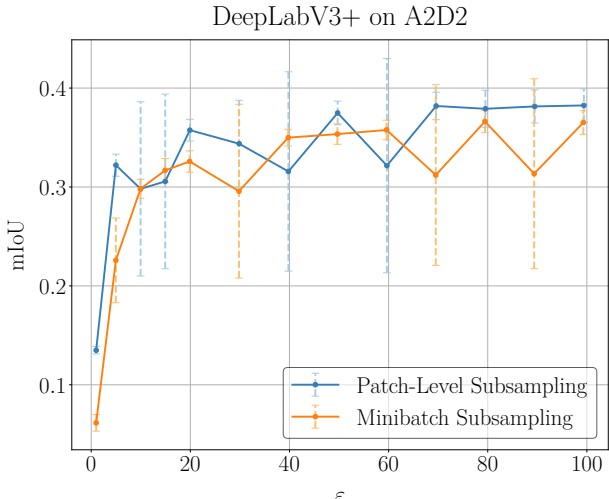

Figure 9: Privacy-utility tradeoff on A2D2 with DeepLabV3+. Results averaged over four seeds, with error bars indicating standard deviation. We assume a private patch size of $10 \times 10$, and set $\delta = 1/18557$.

## C.8 Classification Experiments

In addition to segmentation, we evaluate our method on the image classification task, as mentioned in Section 5.3. In the following, we provide detailed results and discussion.

**Datasets and Models.** We train ResNet-18 (He et al., 2016) and VGG-11 (Simonyan and Zisserman, 2015) on DTD (Cimpoi et al., 2014) and MNIST (LeCun et al., 2002) datasets. DTD dataset has 5640 images of different sizes, with a 1880/1880/1880 train/val/test split. To better align with our method, we drop out the images that have any dimension smaller than 300 and resize the rest to $300 \times 300$ as preprocessing. This transforms the data split to 1879/1878/1880. MNIST has a 60,000/10,000/10,000 train/val/test split consisting of $28 \times 28$ images. For both models, on DTD, we freeze the batch normalization layers (Ioffe and Szegedy, 2015) because they are incompatible with DP-SGD, as mentioned before. Additionally, we initialize the models with ImageNet (Deng et al., 2009) weights, then fine-tune the entire model. On MNIST, we remove the batch normalization layers and initialize randomly.

**Experimental Setup.** For differential privacy, we use the same setup from Section C.3. For hyperparameter configuration, we performed separate tuning for each dataset-model combination. For the DTD dataset, we assumed a private patch size of $5 \times 5$, smaller relative to A2D2 and Cityscapes, as the image sizes are much smaller. All tuning was done with fixed $\varepsilon = 10$ and used for all epsilon values. For ResNet-18 on DTD ($\delta = 1/1879$), the optimal configuration was crop size $100 \times 100$, zero padding, batch size 128, clipping norm $C = 0.5$, learning rate 0.01, and 150 epochs. For VGG-11 on DTD, we changed batch size to 64, learning rate to 0.002, and added weight decay $10^{-4}$, while keeping the rest the same. Unlike the segmentation experiments, we used SGD with momentum 0.9 and a cosine learning rate scheduler for both models. For experiments across privacy budgets, we kept training horizons fixed and adapted the noise multiplier $\sigma$ accordingly. All DTD experiments were repeated with four seeds.

**Limitations with MNIST.** We additionally experimented with MNIST to explore the boundaries of our method's applicability. We used a crop size of $10 \times 10$ and assumed a private patch size of only 1 pixel, as images are much smaller compared to other datasets. However, MNIST presents a fundamental challenge: cropping destroys semantic content regardless of privacy considerations. At $28 \times 28$ resolution, digits contain minimal redundancy, and random crops can make classes not understandable or indistinguishable. For example, cropping "8" can produce shapes similar to "0", "3" or "9". As shown in Table 1, even without privacy noise ($\varepsilon = \infty$), cropping reduces accuracy from over 99% to below 26%. This confirms that our method is unsuitable for datasets where global structure is essential for classification and random cropping fundamentally alters class identity. We include MNIST results to delineate when patch-level privacy analysis

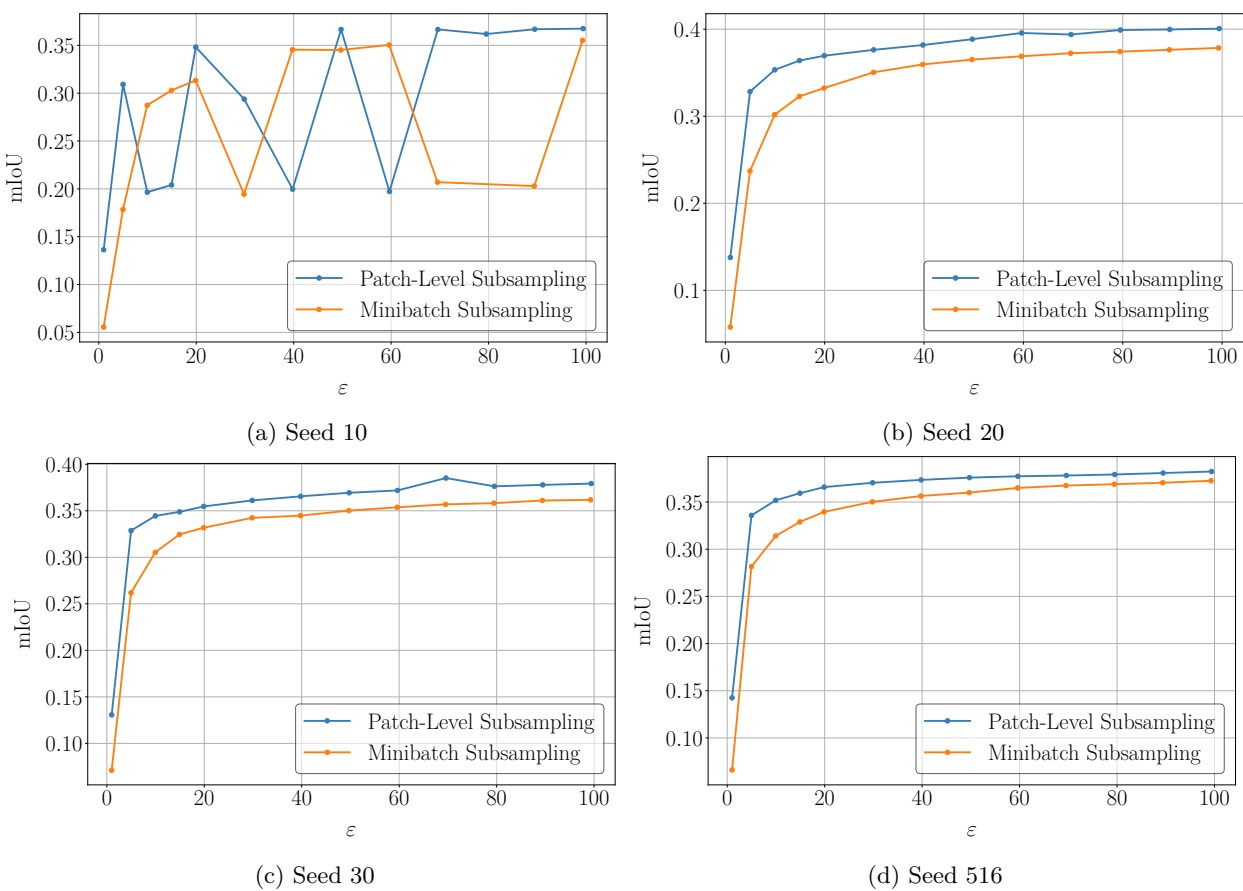

Figure 10: Per-seed privacy-utility tradeoffs for A2D2 with DeepLabV3+. Three seeds converge and confirm the utility benefits of patch-level subsampling. Seed 516 diverged under both patch-level and minibatch subsampling. We assume a private patch size of $10 \times 10$, and set $\delta = 1/18557$.

is *not* appropriate: namely, when the task requires holistic image understanding and cropping cannot preserve semantic content. For MNIST, we removed batch normalization layers entirely and trained from random initialization.

Table 1: MNIST accuracy without differential privacy ($\varepsilon = \infty$). Cropping alone destroys utility.

| Model | No Crop | Crop ($10 \times 10$) |
|-------|---------|------------------------|
| ResNet-18 | 99.4% | 26.0% |
| VGG-11 | 99.2% | 16.7% |

**Results on DTD.** Figure 13 shows privacy-utility tradeoffs for VGG-11 and ResNet-18 on DTD. Consistent with our segmentation experiments, patch-level sampling yields higher accuracy across all tested privacy budgets. For both models, the improvement becomes more pronounced at smaller $\varepsilon$ values, with the largest gains observed in the low to moderate privacy regime. At $\varepsilon = 1$, however, the difference diminishes as the high noise magnitude degrades training for both methods. Per-seed results are provided in Figure 15 and 16.

**Results on MNIST.** As anticipated, both methods perform poorly on MNIST due to the cropping-induced information loss discussed above. Table 1 shows that random cropping alone, even without differential privacy, reduces accuracy from over 99% to below 27%, confirming that cropping destroys class-discriminative information for MNIST. Since cropping already fails in the non-private setting, applying DP-SGD on top of

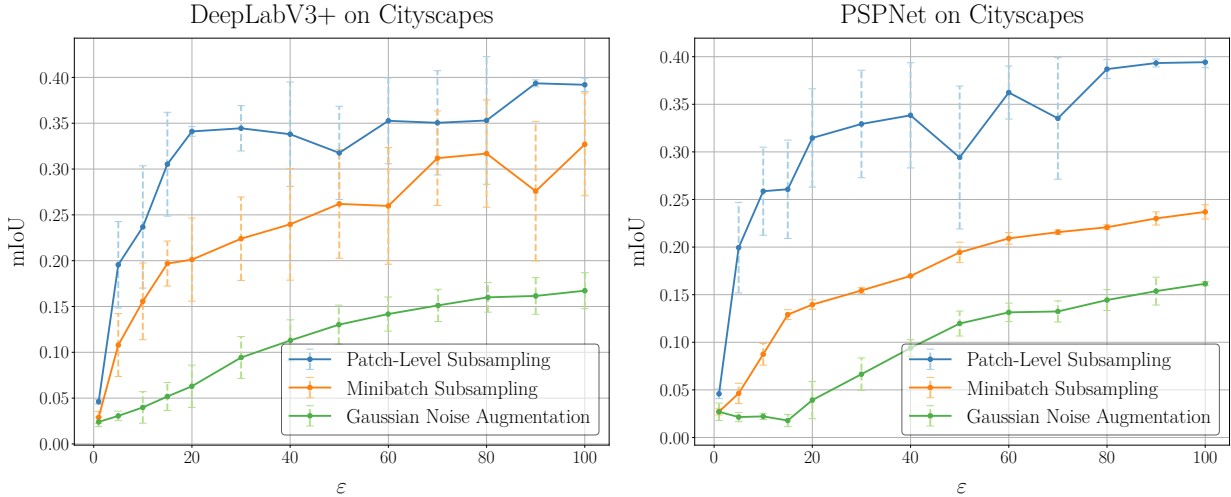

Figure 11: Privacy-utility tradeoff comparing patch-level subsampling (blue), minibatch subsampling (orange), and Gaussian noise augmentation (green) for DeepLabV3+ and PSPNet on Cityscapes. Gaussian noise augmentation performs worse than both alternatives due to the high noise levels required for amplification. $\delta = 1/2975$, private patch size $10 \times 10$.

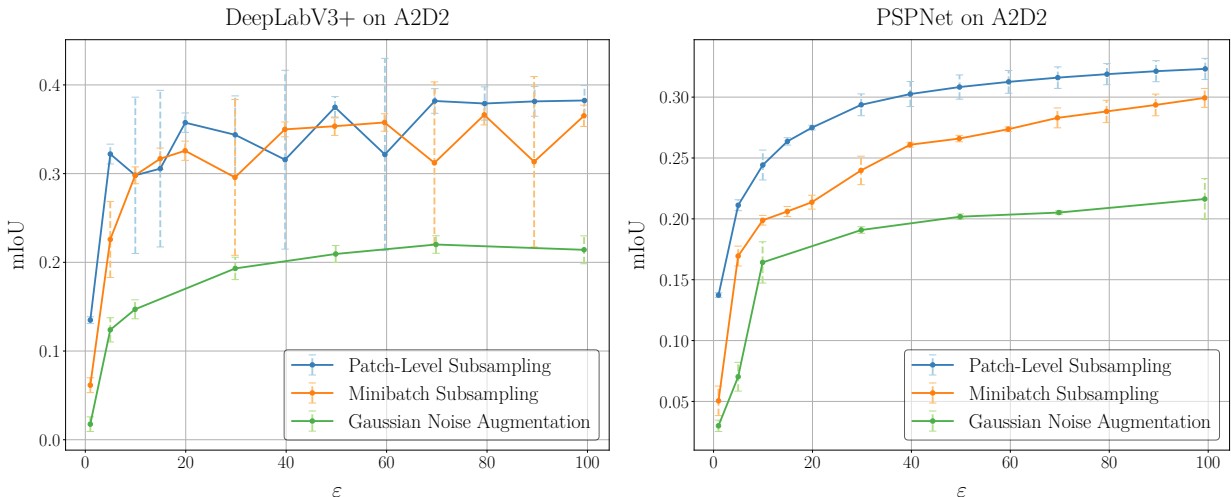

Figure 12: Privacy-utility tradeoff comparing patch-level subsampling (blue), minibatch subsampling (orange), and Gaussian noise augmentation (green) for DeepLabV3+ and PSPNet on A2D2. Gaussian noise augmentation performs worse than both alternatives due to the high noise levels required for amplification. $\delta = 1/18557$, private patch size $10 \times 10$.

it cannot recover utility, as reflected by the divergence and high variance in Figure 14. In such cases, one would not use cropping and instead revert to standard DP-SGD with minibatch subsampling.

# D   Additional Experiments

## D.1   Privacy Profile Experiments

In the main text, we reported privacy profiles for a representative setup. Here, we provide additional variations to demonstrate robustness. We use the hyperparameter values stated in App. C.4.

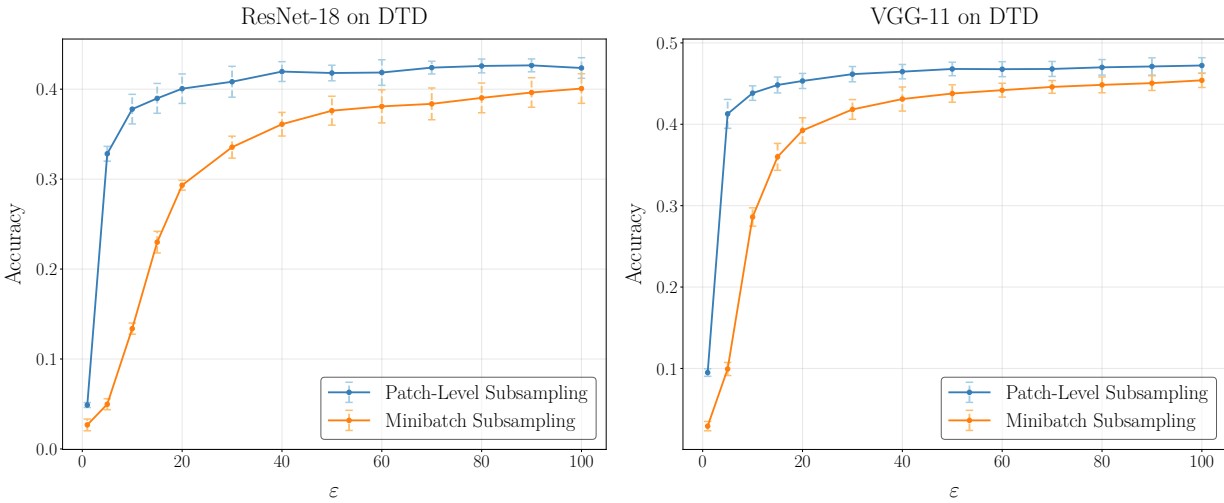

Figure 13: Model performance versus privacy-level $\varepsilon$ for DP-SGD with patch-level sampling (blue) and minibatch subsampling (orange). Results are averaged over four seeds; error bars show standard deviation. We use $\delta = 1/1879$, following $\delta = 1/\texttt{epoch\_size}$ and a private patch size of $5 \times 5$ is assumed. DG-SGD with patch-level privacy overperforms significantly, given the exact same setup.

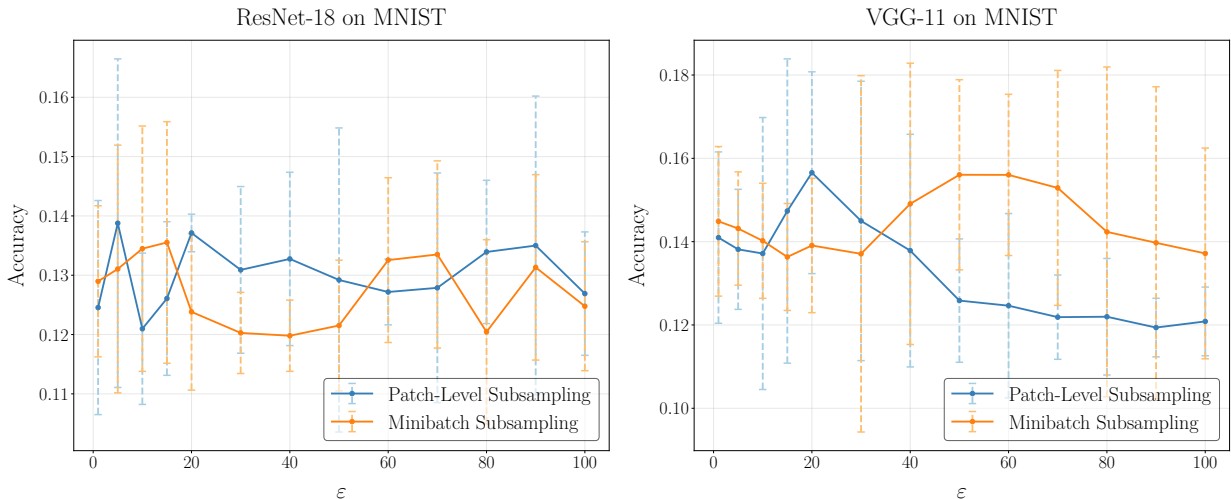

Figure 14: Model performance versus privacy-level $\varepsilon$ for DP-SGD with patch-level sampling (blue) and minibatch subsampling (orange). Results are averaged over four seeds; error bars show standard deviation. We use $\delta = 1/60000$, following $\delta = 1/\texttt{epoch\_size}$ and a private patch size of $5 \times 5$ is assumed.

### D.1.1 Varying Noise Levels

We vary the Gaussian noise multiplier across $\sigma \in \{1, 2, 4\}$, fixing crop size $100 \times 100$, and private patch size $10 \times 10$. For the data-level noise baseline, we set $\sigma_{\text{data}} = 1000$. As expected, we observe in Fig. 17 that larger noise multipliers reduce privacy leakage, shifting the profiles downward. Notably, while the data-level baseline requires extreme values such as $\sigma_{\text{data}} = 1000$ to achieve modest privacy protection, comparably small increases in the multiplier for patch-level subsampling yield meaningful improvements.

### D.1.2 Varying Private Patch Size

We next analyze the effect of enlarging the private region itself. Fixing crop size to $100 \times 100$ and Gaussian noise multiplier $\sigma = 1$, we vary the private patch side length across $\{10, 20, 40\}$, while keeping the data-level

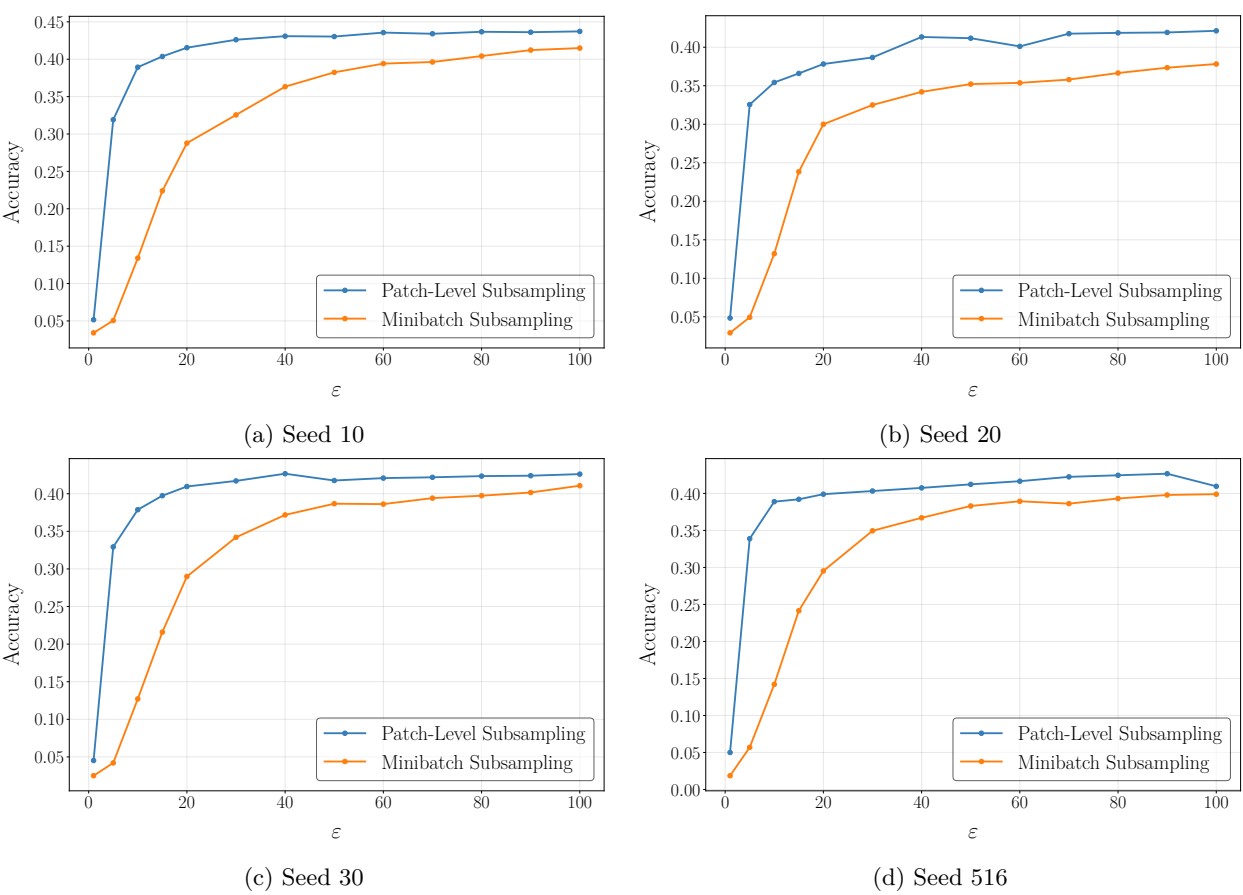

Figure 15: Per-seed privacy-utility tradeoffs of ResNet-18 on DTD. We assume a private patch size of $5 \times 5$, and set $\delta = 1/1879$.

baseline constant at $\sigma_{\mathrm{data}} = 1000$. As shown in Fig. 18, larger private patches gradually bring patch-level subsampling closer to the behavior of standard minibatch sampling, since the intersection probability between the crop and the private region increases. This reduces the effective amplification and slowly narrows the gap between the two methods. Additionally, the negative effect of enlarging private patches is even more pronounced for the data-level baseline, where privacy leakage grows substantially despite already extreme noise levels.

### D.1.3 Varying Crop Sizes

We now examine how the choice of crop size influences the privacy profiles. Fixing private patch size to $20 \times 20$ and Gaussian noise multiplier $\sigma = 2$, we vary the crop side length across $\{100, 200, 400\}$ pixels. For the data-level baseline, we again fix $\sigma_{\mathrm{data}} = 1000$. As shown in Fig. 19, larger crop sizes weaken the benefits of patch-level subsampling, and its privacy profiles shift upward toward the minibatch sampling privacy profile as crop size grows. The effect and its reasoning are very similar to those of private patch size, but it has zero effect on both of the baseline privacy profiles.

### D.1.4 Varying Image Resolution

We now analyze the effect of changing the overall image resolution while keeping other parameters fixed. We compare experiments with image sizes of $1000 \times 1000$ and $2000 \times 2000$, fixing crop size to $200 \times 200$, private patch size to $20 \times 20$, and Gaussian noise multiplier $\sigma = 2$. The data-level baseline is again set to $\sigma_{\mathrm{data}} = 1000$. As shown in Fig. 20, increasing the image resolution strengthens the relative amplification

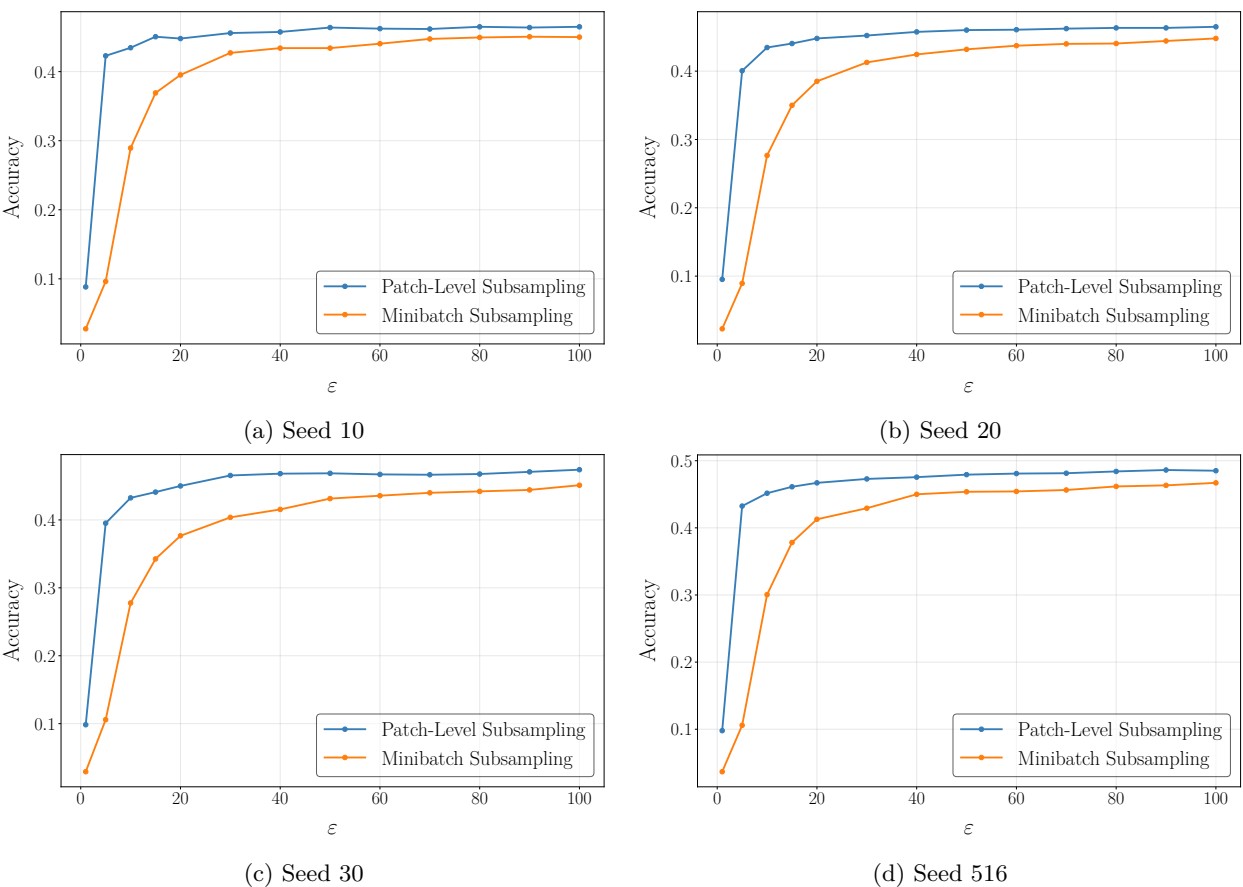

Figure 16: Per-seed privacy-utility tradeoffs of VGG-11 on DTD. We assume a private patch size of $5 \times 5$, and set $\delta = 1/1879$.

of patch-level subsampling. The effect and reasoning behind it are the same as in crop size, just for the variation in the opposite direction. An increase in the image size affects the privacy profile of patch-level subsampling positively, whereas an increase in the crop size affects it negatively.

### D.1.5 Varying Data-Level Noise

We finally turn to the data-level noise addition baseline and vary the injected noise standard deviation directly. Fixing crop size to $200 \times 200$, private patch size to $20 \times 20$, Gaussian noise multiplier $\sigma = 2$, and image resolution $1000 \times 1000$, we compare $\sigma_{\text{data}} \in \{1000, 2000\}$. As shown in Fig. 21, even doubling the standard deviation of pixel-level Gaussian noise fails to match the privacy guarantees achieved by patch-level subsampling in this setup.

### D.1.6 Varying Private Patch Geometry

In the main text and experiments above, we assume square private patches. Here, we demonstrate that our framework generalizes to arbitrary shapes, as discussed in Section 4.4. While closed-form expressions exist for rectangles (and squares) (Lemma 1), inclusion probabilities for other shapes can be computed by iterating over all crop positions. We compare privacy profiles for three geometries: (1) rectangular patches with varying aspect ratios, (2) circular regions (e.g., approximating faces), and (3) irregular blobs (cluster of 2 circles). Results are shown in Figure 22. These results confirm that our theoretical framework extends beyond square patches, and practitioners can use domain-appropriate shapes for tighter analysis.

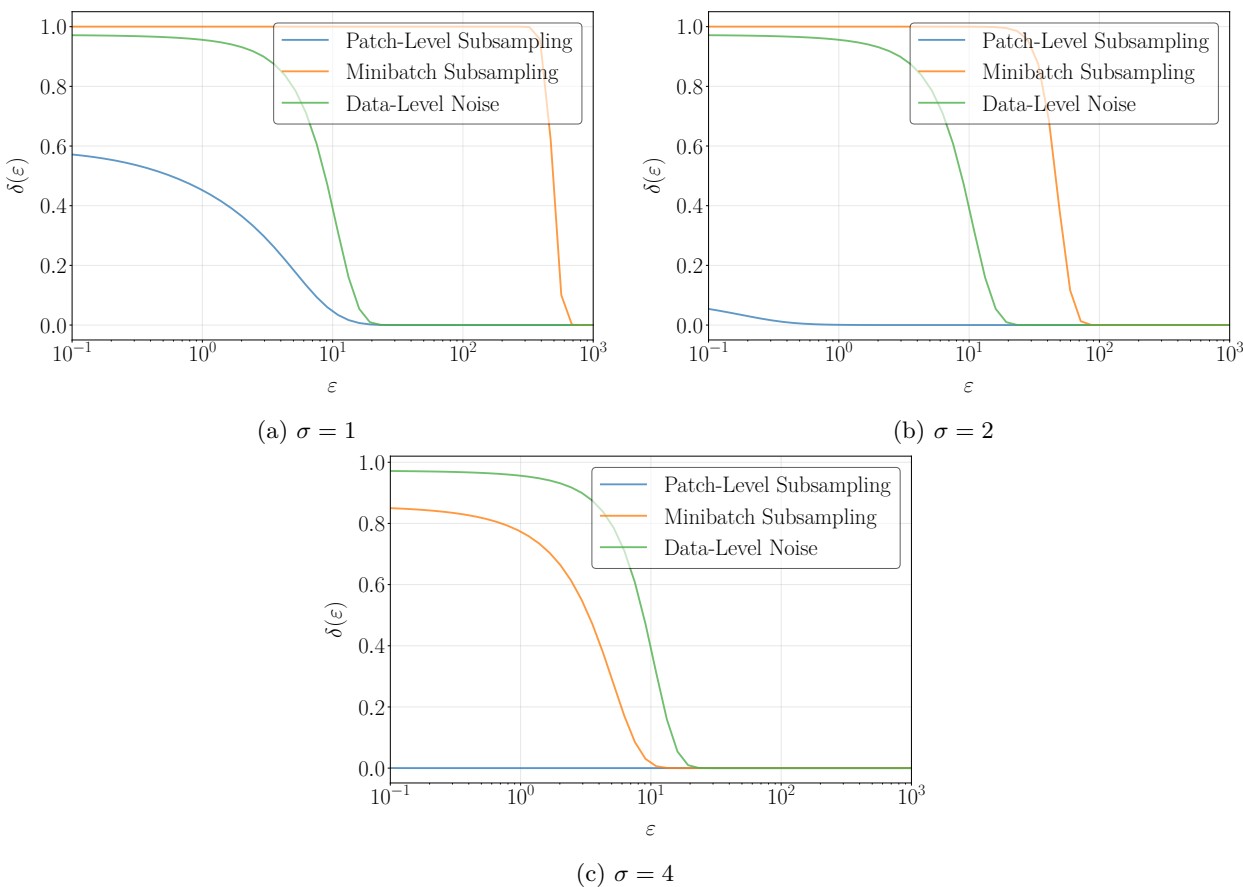

Figure 17: Privacy profiles for different mechanisms: DP-SGD with patch-level subsampling (blue), standard minibatch subsampling (orange), and data-level noise addition (green). We vary the noise levels $\sigma$ for the subsampling mechanisms, while keeping the data-level baseline fixed at $\sigma_{\text{data}} = 1000$.

## D.2 Additional Experiments on Crop and Patch Size

In Section 5.2, we analyzed how the privacy parameter $\varepsilon$ varies with crop size and private patch size, highlighting the role of intersection probability in determining the strength of patch-level amplification. There, we reported results for a representative setting with $\sigma \in \{4.0, 4.5, 5.0\}$. Here, we expand these experiments by systematically varying the Gaussian noise multiplier and repeating the sweeps across two image resolutions. As before, we fix `epoch_size` $= 10^5$, and $\delta = 1/$`epoch_size` and perform two sweeps in each configuration: (1) varying crop sizes with private patch size fixed at $10 \times 10$, and (2) fixing crop size at $450 \times 450$ while varying private patch sizes. Together with the sensitivity analyses in Appendix D.1, these results confirm that our findings are robust across a wide range of settings.

We vary the Gaussian noise multiplier between 1.0 and 5.0 in steps of 0.5, and repeat both sweeps at two image resolutions: $1000 \times 1000$ (Figures 23 - 25) and $1000 \times 2000$ (Figures 26 - 28). As expected, larger noise multipliers uniformly lower the $\varepsilon$ curves, while the qualitative dependence on crop and patch size remains unchanged. For the crop-size sweeps, $\varepsilon$ rises rapidly with increasing crop size and saturates once intersection probability reaches 1, at which point patch-level subsampling collapses to the minibatch baseline. For the patch-size sweeps, $\varepsilon$ grows approximately linearly with the side length of the private region, again approaching the minibatch baseline at saturation. Importantly, the rectangular image resolution illustrates that saturation is not guaranteed, even with very large crops or patches. Depending on the image dimensions, the intersection probability may never reach 1, and patch-level subsampling preserves an advantage throughout. This reinforces that dataset geometry itself can modulate the strength of amplification.

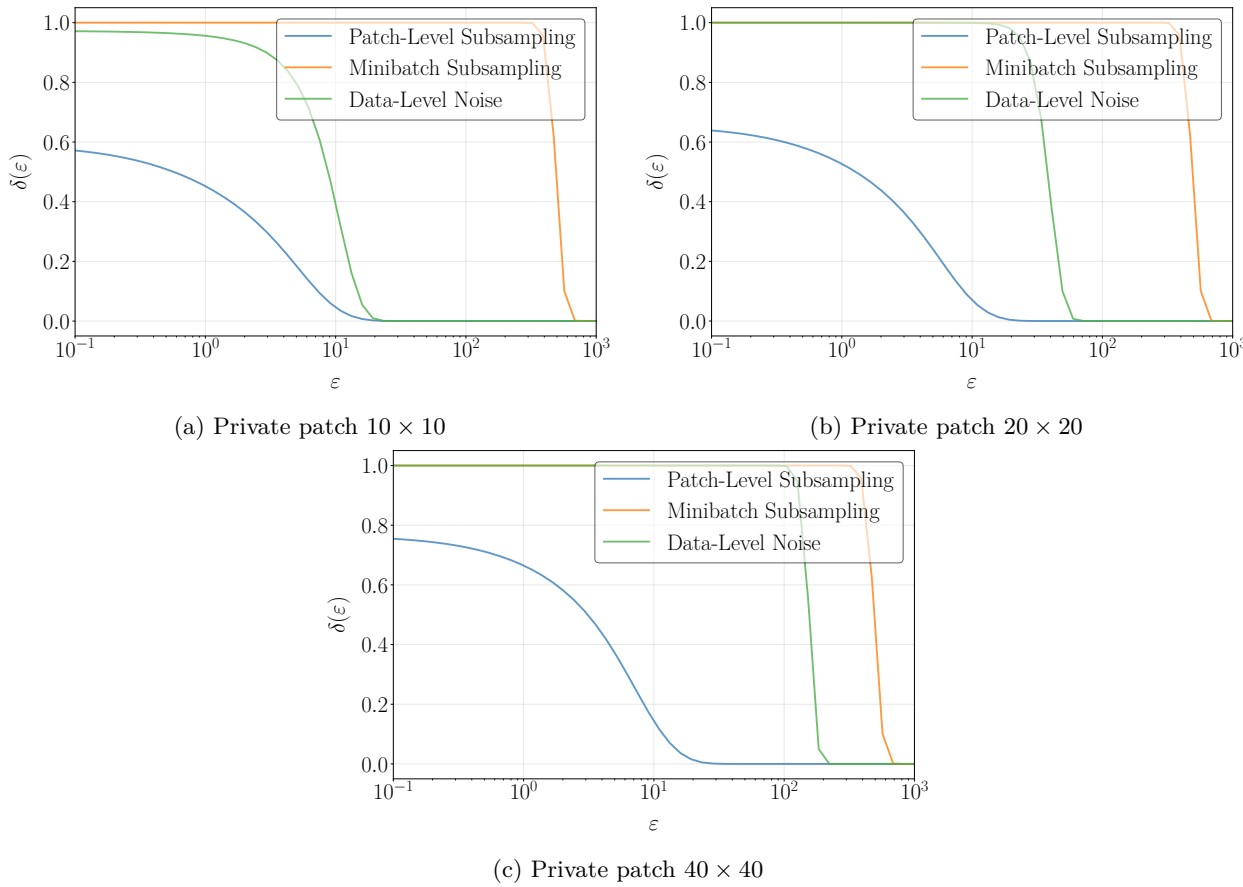

(a) Private patch $10 \times 10$  (b) Private patch $20 \times 20$

(c) Private patch $40 \times 40$

Figure 18: Privacy profiles for different mechanisms under varying private patch sizes: DP-SGD with patch-level subsampling (blue) and standard minibatch subsampling (orange) with identical $\sigma = 1$, and data-level noise addition (green) with $\sigma_{\texttt{data}} = 1000$.

## D.3 Influence of Padding on Privacy

Finally, we investigate how image padding affects the strength of patch-level amplification. Padding alters the effective geometry of the cropping operation by adding empty pixels along the edges of the image before cropping. Therefore, it modulates the intersection probability between the private region and the sampled crop.

Unless otherwise stated, all experiments use images of size $1000 \times 1000$, clipping norm $C = 2.0$, batch size 100, epoch size 3000, and 100 training epochs. We fix $\delta = 1/\texttt{epoch\_size}$ and assume $\texttt{epoch\_size} = 10^5$ throughout. On top of these defaults, we perform three sweeps under varying padding configurations: (1) varying crop side lengths with private patch size fixed at $10 \times 10$ and Gaussian noise multipliers $\sigma \in \{4.0, 4.5, 5.0\}$, (2) varying private patch sizes with crop size fixed at $500 \times 500$ and the same set of noise multipliers, and (3) varying Gaussian noise multipliers with crop size fixed at $500 \times 500$ and private patch size $20 \times 20$. In each case, we repeat the experiment under multiple padding values to assess their influence on $\varepsilon$. Figures 29–31 show the results.

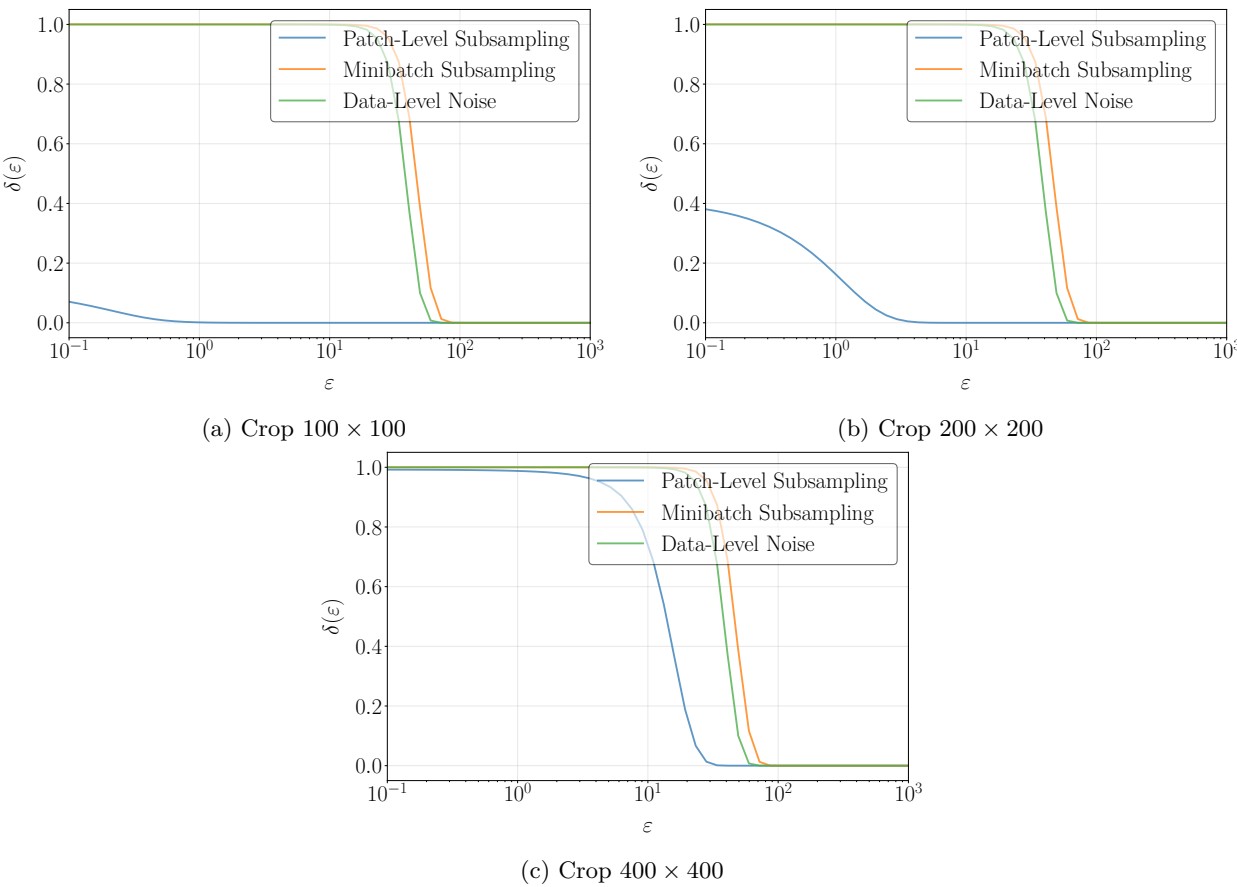

Figure 19: Privacy profiles for different mechanisms under varying crop sizes: DP-SGD with patch-level subsampling (blue) and standard minibatch subsampling (orange) with identical $\sigma = 2$, and data-level noise addition (green) with $\sigma_{\mathtt{data}} = 1000$.

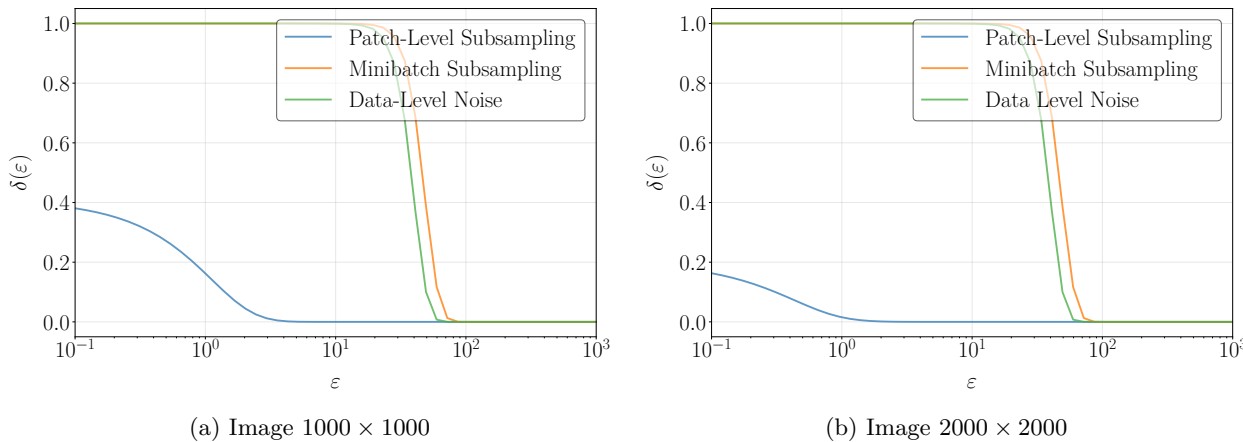

Figure 20: Privacy profiles for different mechanisms under varying image resolutions: DP-SGD with patch-level subsampling (blue), standard minibatch subsampling (orange), and data-level noise addition (green). Crop size $200 \times 200$, private patch $20 \times 20$, $\sigma = 2$, $\sigma_{\mathtt{data}} = 1000$.

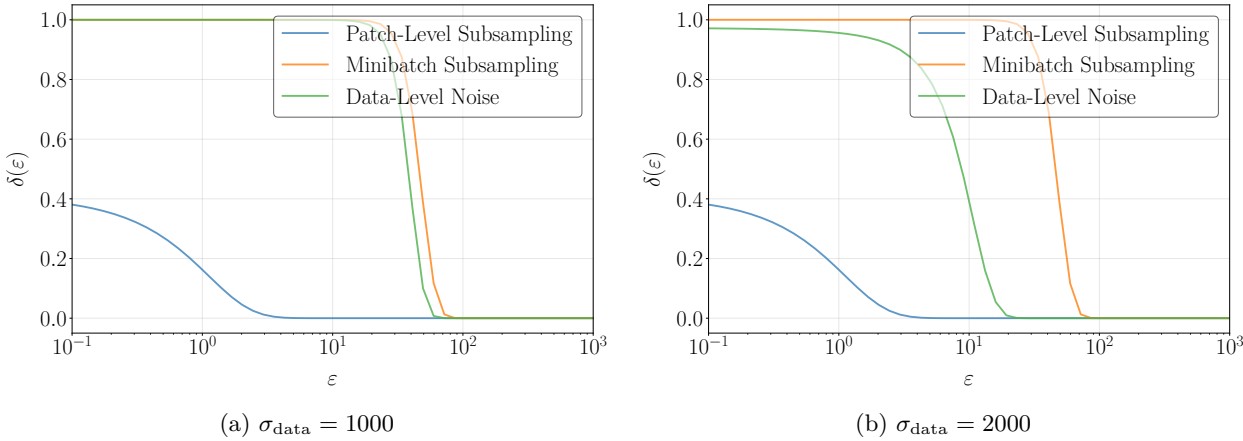

(a) $\sigma_{\mathrm{data}} = 1000$
(b) $\sigma_{\mathrm{data}} = 2000$

Figure 21: Privacy profiles for different mechanisms under varying data-level noise standard deviations: DP-SGD with patch-level subsampling (blue), standard minibatch subsampling (orange), and data-level noise addition (green). Crop size $200 \times 200$, private patch $20 \times 20$, $\sigma = 2$.

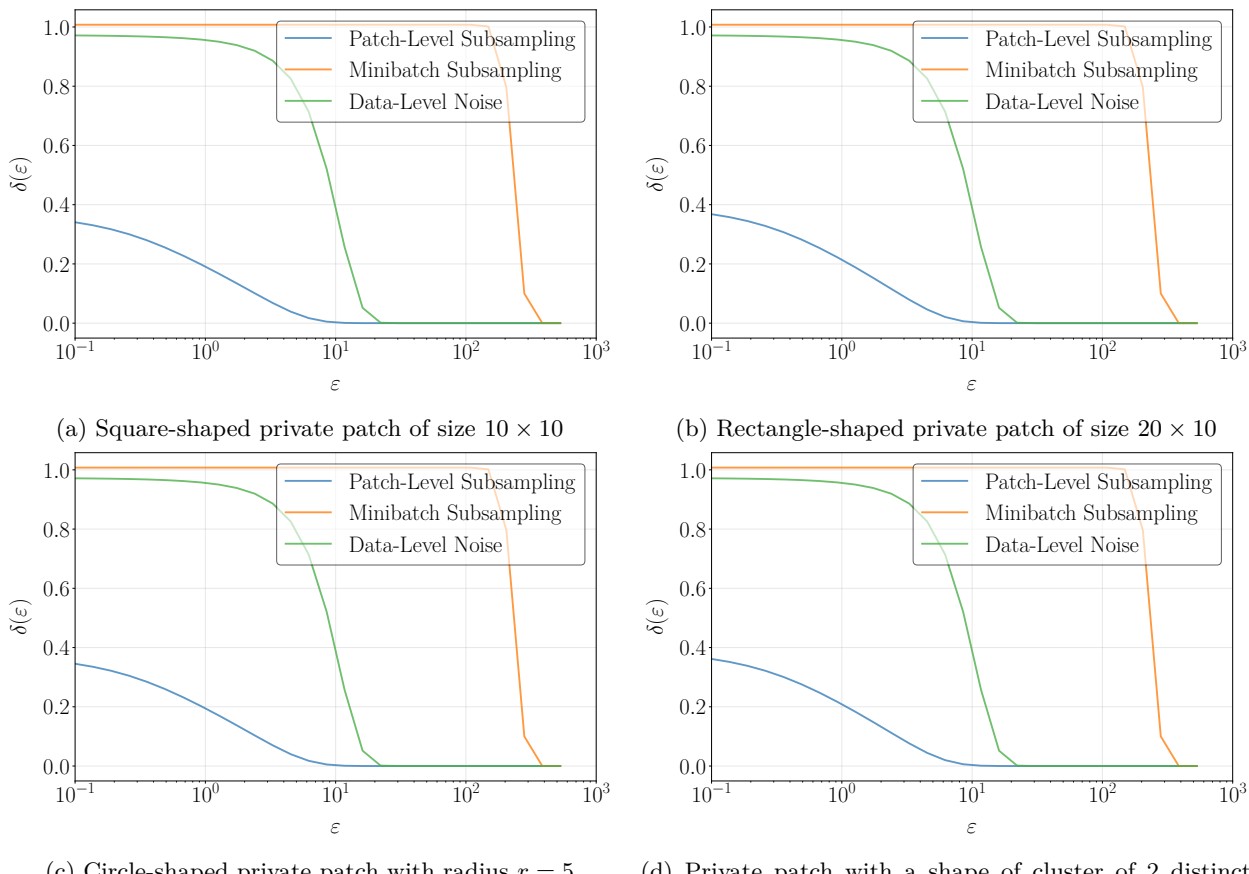

(a) Square-shaped private patch of size $10 \times 10$

(b) Rectangle-shaped private patch of size $20 \times 10$

(c) Circle-shaped private patch with radius $r = 5$

(d) Private patch with a shape of cluster of 2 distinct circles, each circle with radius $r = 4$ and a gap in between of 2 pixels

Figure 22: Privacy profiles for different mechanisms: DP-SGD with patch-level subsampling (blue), standard minibatch subsampling (orange), and data-level noise addition (green). We vary the shapes of the private patches.

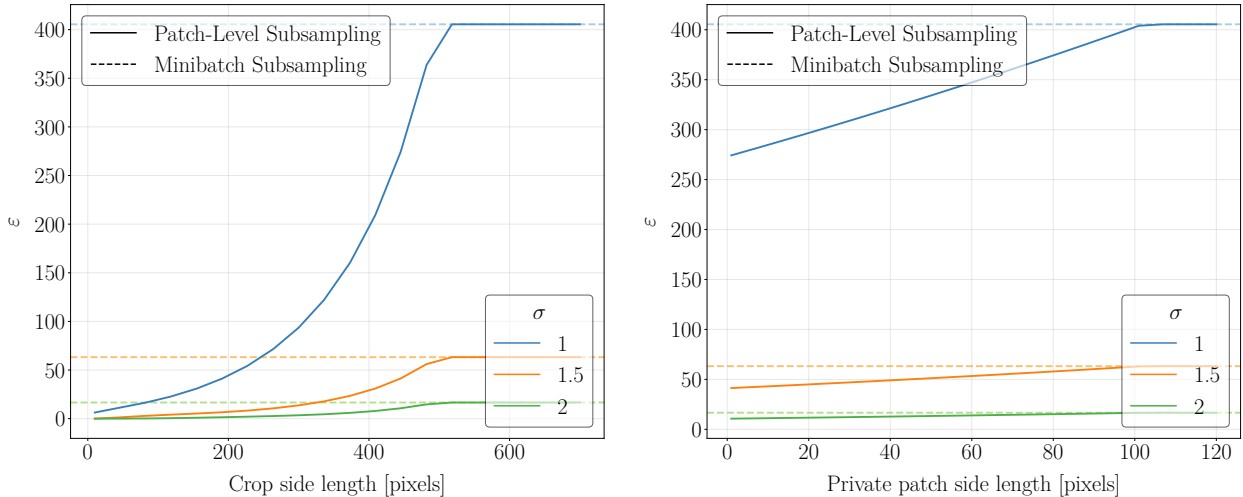

Figure 23: $\varepsilon$ as a function of crop size (left) and private patch size (right), at $\delta = 10^{-5}$ and image resolution $1000 \times 1000$ for $\sigma \in \{1.0, 1.5, 2.0\}$.

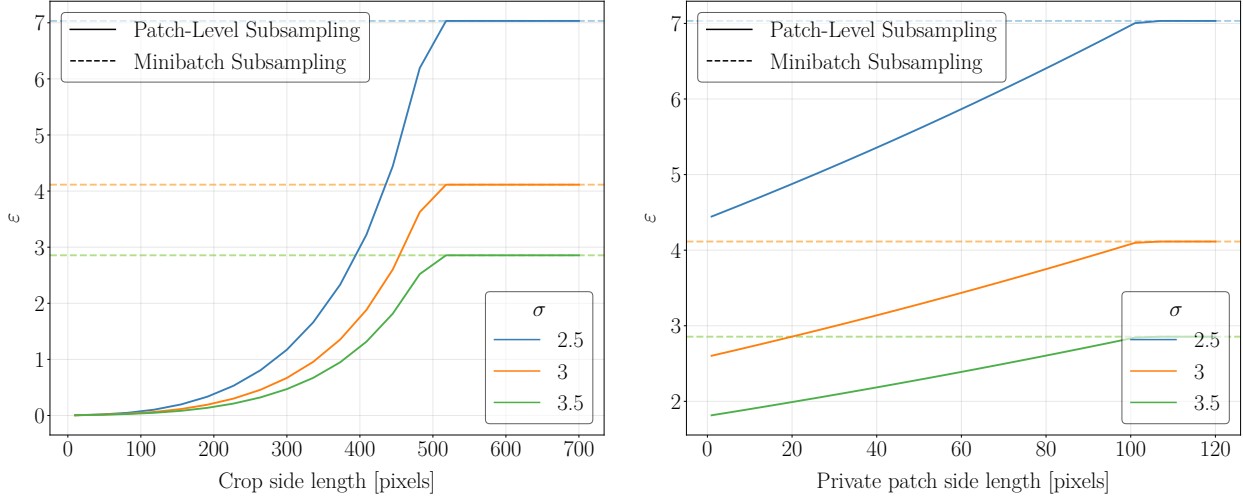

Figure 24: $\varepsilon$ as a function of crop size (left) and private patch size (right), at $\delta = 10^{-5}$ and image resolution $1000 \times 1000$ for $\sigma \in \{2.5, 3.0, 3.5\}$.

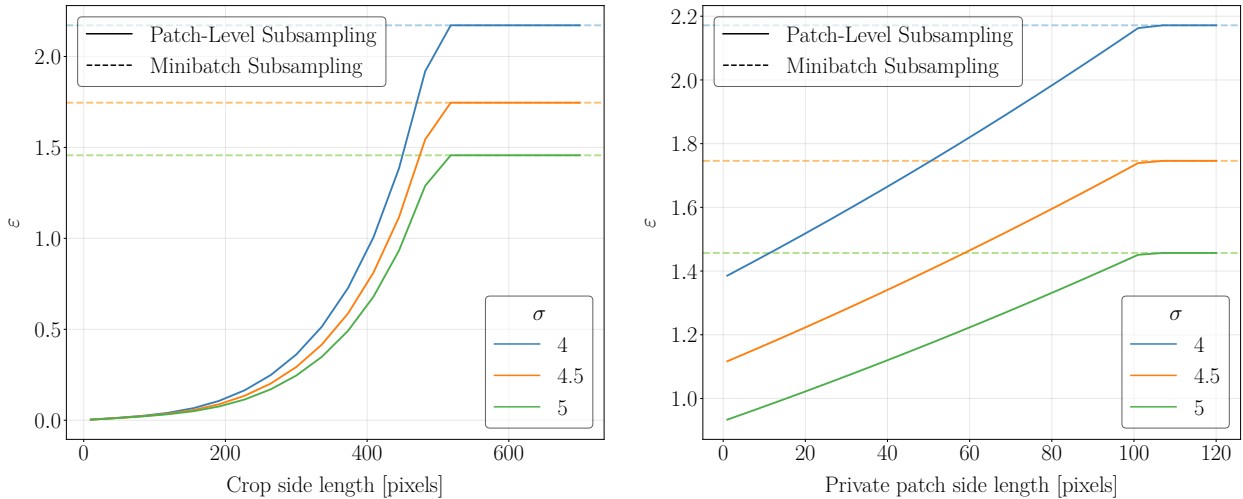

Figure 25: $\varepsilon$ as a function of crop size (left) and private patch size (right), at $\delta = 10^{-5}$ and image resolution $1000 \times 1000$ for $\sigma \in \{4.0, 4.5, 5.0\}$.

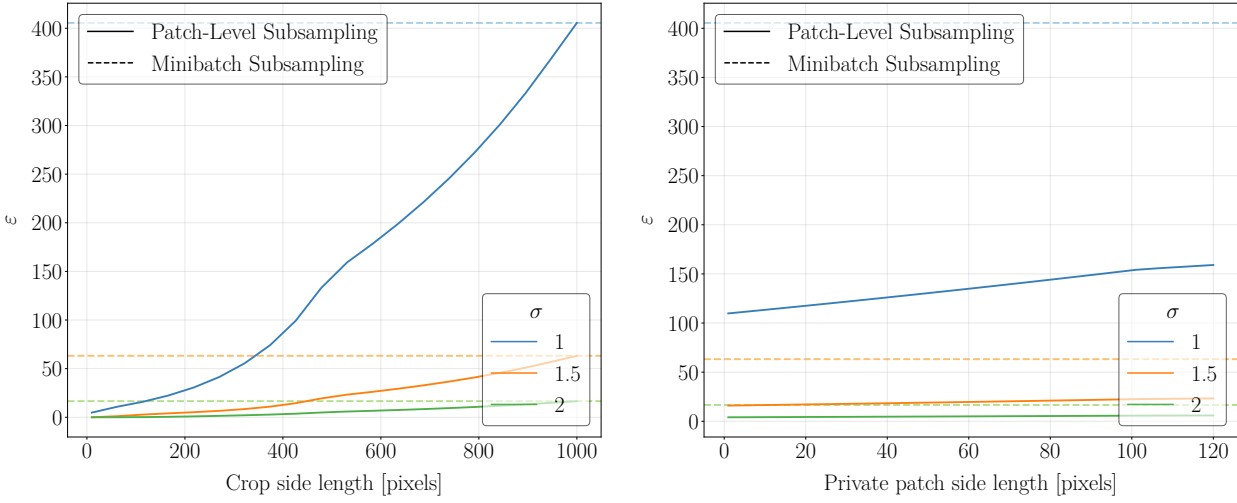

Figure 26: $\varepsilon$ as a function of crop size (left) and private patch size (right), at $\delta = 10^{-5}$ and image resolution $1000 \times 2000$ for $\sigma \in \{1.0, 1.5, 2.0\}$.

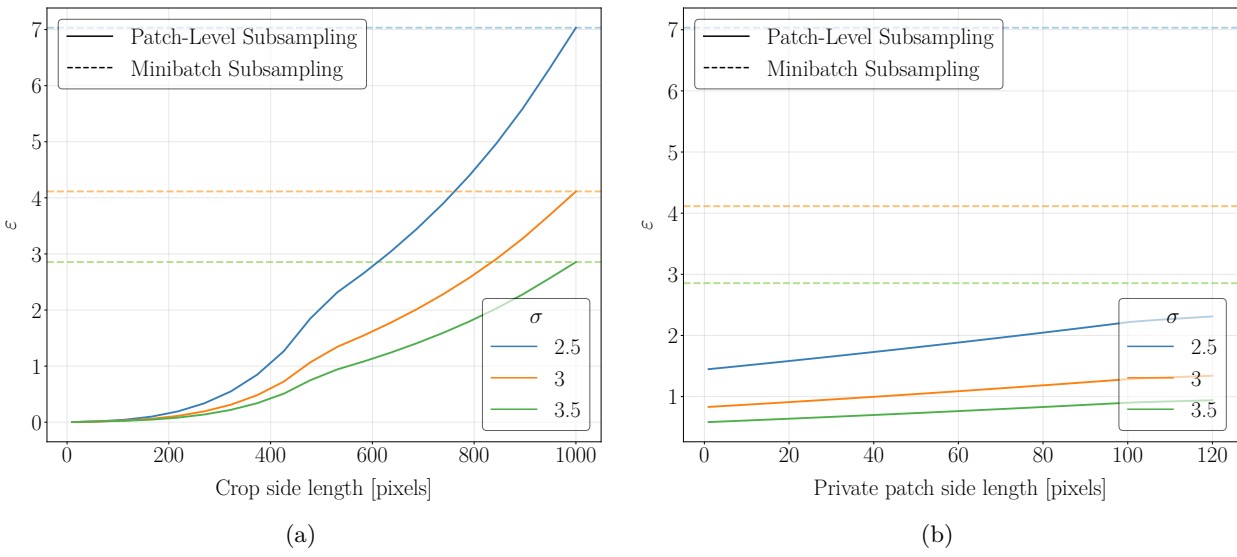

Figure 27: $\varepsilon$ as a function of crop size (left) and private patch size (right), at $\delta = 10^{-5}$ and image resolution $1000 \times 2000$ for $\sigma \in \{2.5, 3.0, 3.5\}$.

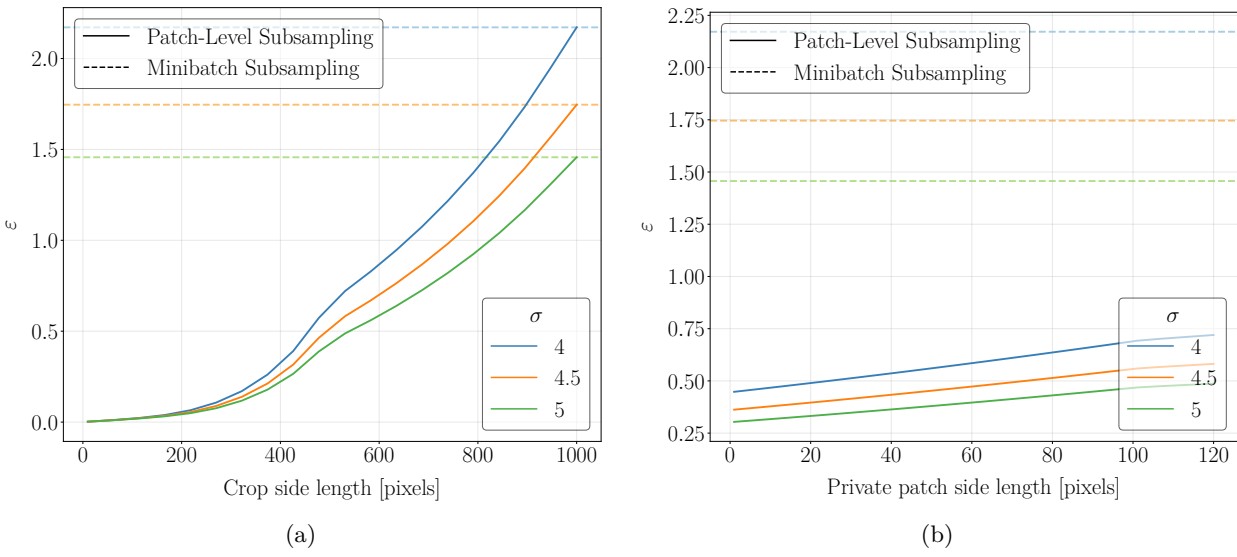

Figure 28: $\varepsilon$ as a function of crop size (left) and private patch size (right), at $\delta = 10^{-5}$ and image resolution $1000 \times 2000$ for $\sigma \in \{4.0, 4.5, 5.0\}$.

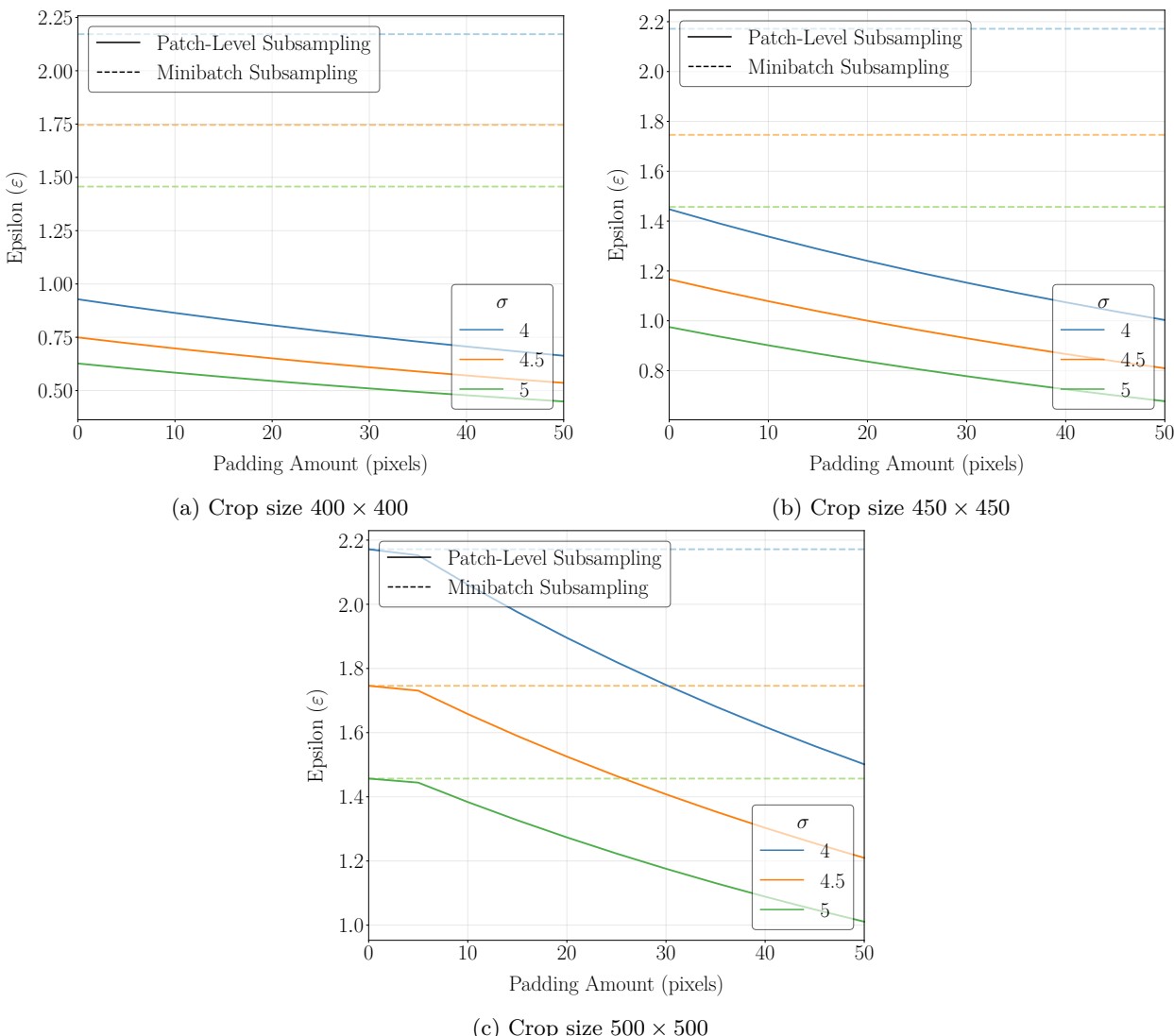

(a) Crop size $400 \times 400$

(b) Crop size $450 \times 450$

(c) Crop size $500 \times 500$

Figure 29: $\varepsilon$ vs. padding amount for crop sizes (a) $400 \times 400$, (b) $450 \times 450$, (c) $500 \times 500$, with patch size $10 \times 10$ and noise multipliers $\sigma \in \{4.0, 4.5, 5.0\}$.

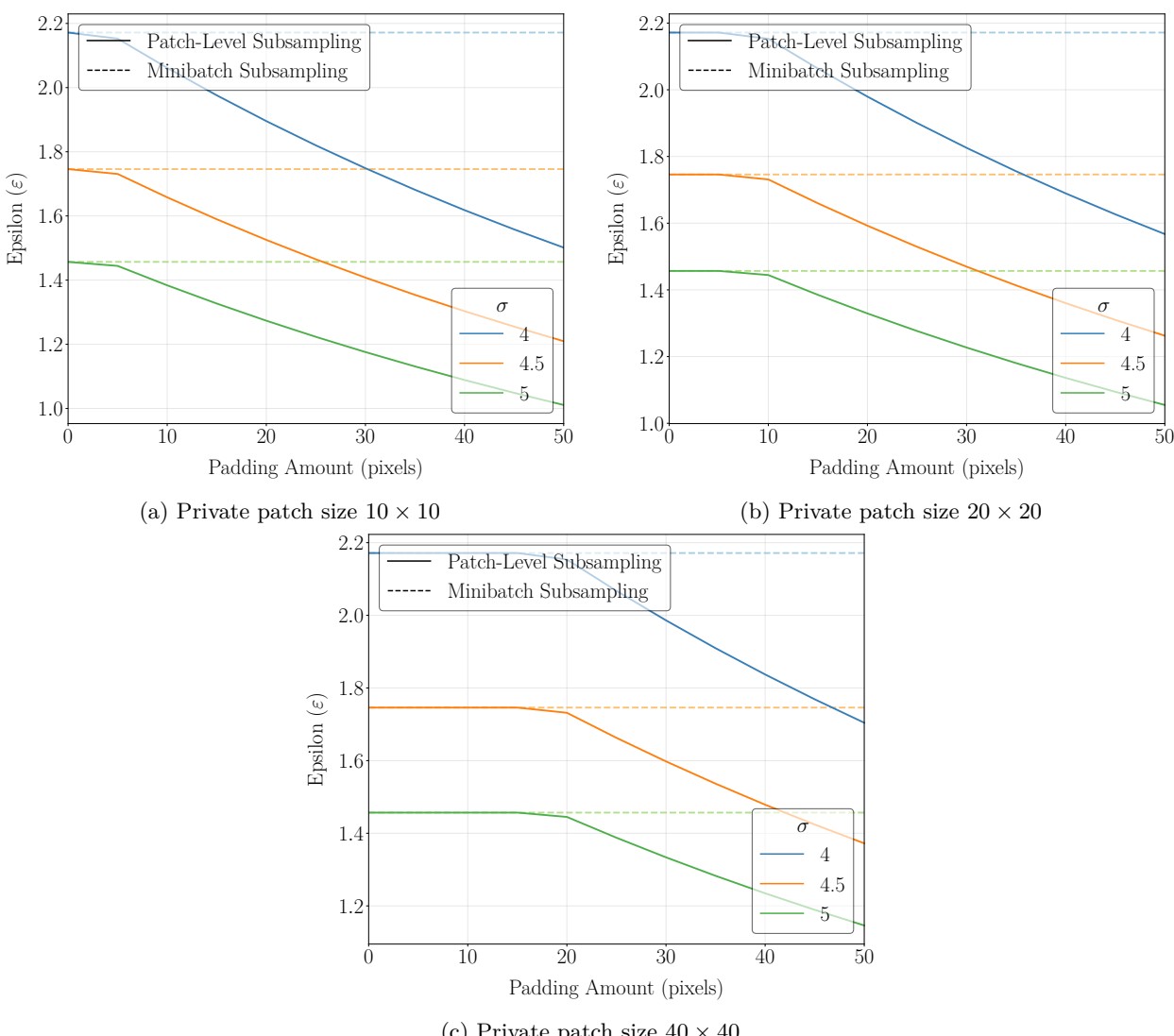

(a) Private patch size $10 \times 10$       (b) Private patch size $20 \times 20$

(c) Private patch size $40 \times 40$

Figure 30: $\varepsilon$ vs. padding amount for private patch sizes (a) $10 \times 10$, (b) $20 \times 20$, (c) $40 \times 40$, with crop size $500 \times 500$ and noise multipliers $\sigma \in \{4.0, 4.5, 5.0\}$.

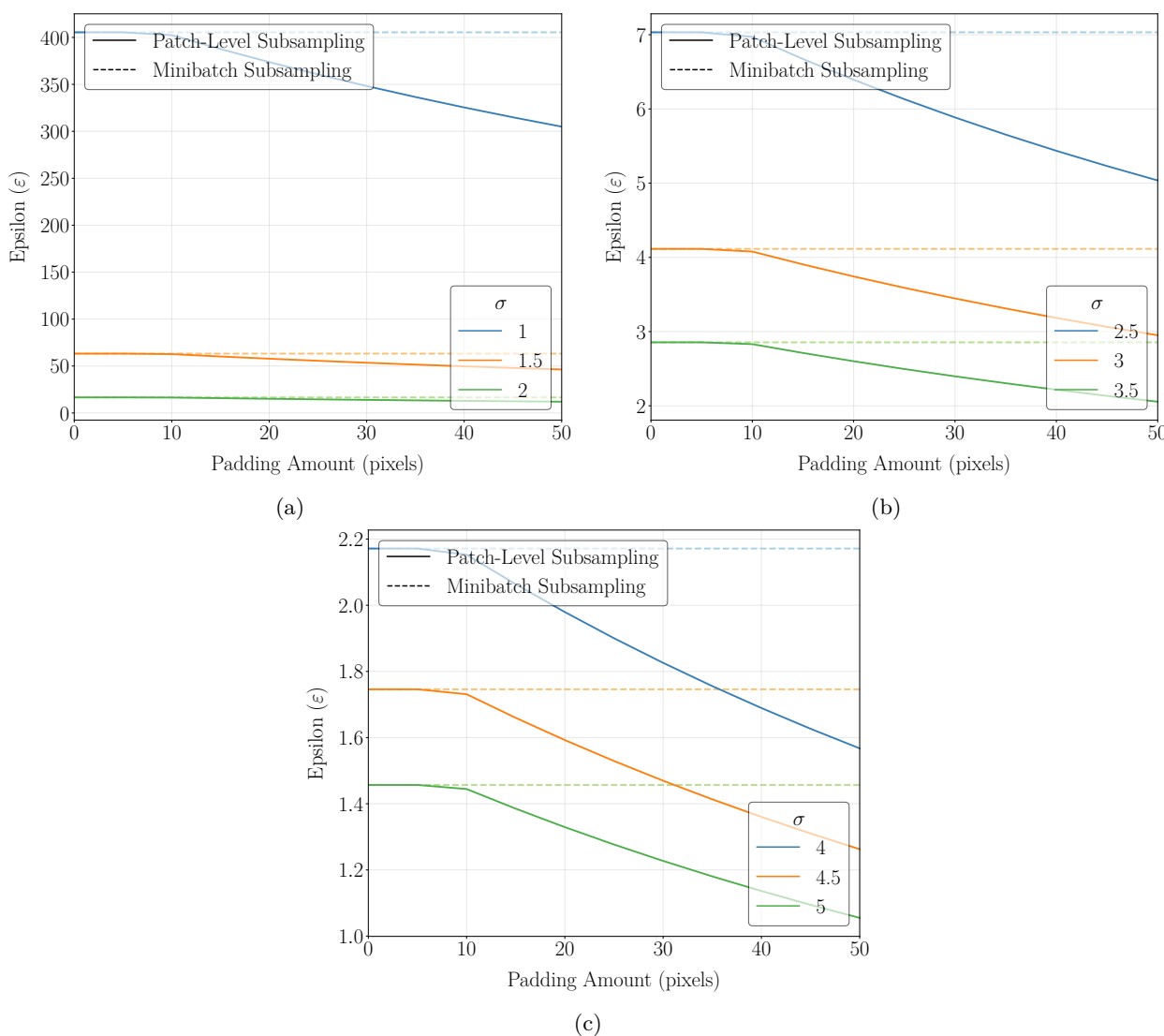

Figure 31: $\varepsilon$ vs. padding amount for noise values (a) $\sigma \in \{1.0, 1.5, 2.0\}$, (b) $\sigma \in \{2.5, 3.0, 3.5\}$, (c) $\sigma \in \{4.0, 4.5, 5.0\}$, with crop size $500 \times 500$ and private patch size $20 \times 20$.

