# OpenReview forum: "Amplified Patch-Level Differential Privacy for Free via Random Cropping"
_TMLR — Accepted by TMLR_

### Review · Reviewer_HPaG · 2025-11-19

**Summary Of Contributions:**

In this paper, Amplified Patch-Level Differential Privacy for Free via Random Cropping, authors propose a principled way to get privacy amplification in vision DP-SGD by using randomness from random cropping. The privacy gain comes without having to change model architecture or training procedure. They introduce a patch-level neighboring relation, where two datasets differ only within a bounded region of a single image. This additional sampling effect, when composed with standard minibatch subsampling, leads to tighter privacy bounds and improved model utility without modifying the training pipeline. The work is promising but needs a major revision.

Strengths:

•	Introduction reads well and the motivation is described clearly

•	Strong theoretical link between random cropping, a data augmentation technique, and privacy amplification

•	Treating random cropping as a source of privacy amplification

•	The privacy amplification is achieved “for free,” and without modifying the training pipeline

Weaknesses:

•	The evaluation is only limited to semantic segmentation

**Audience:**

Yes

**Audience Explanation:**

The paper is about differential privacy in deep learning, a core topic for TMLR. There are many previously published papers in TMLR, addressing and improving differential privacy in different settings.

**Broader Impact Concerns:**

The paper introduces patch-level DP, which is weaker than standard record-level DP, and this distinction may be misunderstood. The method also assumes a fixed known sensitive region, and unknown private regions or misidentifying them could hurt the actual privacy. The approach might be capable of training better performing models under relaxed privacy notions, so there is potential for misuse.

**Claims And Evidence:**

No

**Claims Explanation:**

The proposed claims are broad. All empirical evaluations are limited to semantic segmentation, but the method is claimed to be broadly applicable across vision tasks. Some potential problems in simpler classification tasks (like MNIST) are briefly mentioned, but no further discussions or experiments are presented to validate whether the amplification benefits persist in other settings.

**Requested Changes:**

Critical comments:

• Consider adding a discussion on how patch-level DP relates to normal (record-level) DP and when the former is the right guarantee for the application.

• The paper assumes a known private patch size. Please discuss how misidentifying or unknown private regions could reduce protection that does not affect the privacy notion here. Additionally, the real data might involve multiple sensitive regions. A discussion and analysis should be added.

• All reported privacy–utility tradeoffs are on semantic segmentation tasks. Since the paper itself notes random cropping can hurt performance in simpler datasets such as MNIST, please include results on a classification task. In order to show the privacy improvement generalizes beyond segmentation.


Suggestions:

• Definition 3 states “x_i(s,t)\neq x_i^\prime(s,t) for some (∃ vs. ∀)  (s,t)\in R.”
Please confirm and discuss that the intent is an unbounded in-region substitution. In other words, outside R, the images are identical, while inside R, any modification, including a single-pixel change, constitutes adjacency. And the entire R region is not necessarily privatized, as demonstrated in Figure 1.

• δ choices are inconsistent across sections and need unification. Both δ=10⁻⁵ and δ=1/epoch_size have been used.

• Reproducibility of the results is a concern. Results on Cityscapes have high error bars, at some points, minibatch subsampling showing better performance. Please discuss this in section 5.3.

• In the privacy-utility figure captions, please explicitly state δ and patch size.

---

> ### Author Response · Authors · 2026-01-02
> **Rebuttal by Authors - part 1**
>
> Thank you for your review and for recognizing the strong theoretical link between random cropping and privacy amplification. Based on your valuable feedback, we have updated our manuscript. For your convenience, all changes are highlighted in blue, and a summary of important changes is provided in our global rebuttal comment. We address all your concerns below.
>
> ## W1 & RC3. Generalization to other vision tasks
>
> Our privacy amplification is **task-agnostic**: it improves privacy accounting for DP-SGD whenever random cropping is used, independent of the task. For a given noise level and crop size, our analysis yields strictly stronger privacy guarantees when the crop can exclude the private patch (see Figure 2 and Figures 16-21). In the worst case, where the crop always includes the private patch, it recovers standard DP-SGD exactly. This is a property of the accounting, not the learning objective.
>
> We agree that additional experiments strengthen this claim. We have added classification experiments using ResNet-18 and VGG-11 on DTD and MNIST (see the updated Section 5.3 and Appendix C.8, highlighted in blue).
>
> Our DTD results confirm that the improved privacy-utility tradeoffs clearly generalize to classification (see Figure 12).
>
> On MNIST, cropping substantially harms utility, even at $\varepsilon = \infty$ (see Table 1); however, this does not contradict our claims, as our method still provides stronger privacy guarantees. However, these stronger privacy guarantees do not translate to better performance because cropping already destroys the semantic information; at that point, the noise has no additional effect. We stress that we deliberately chose a fair evaluation setup and include also MNIST results to show when the practical benefit is limited (see Figure 13). Additionally, we have further highlighted this in our discussion of limitations (see Section 6, paragraph 4).
>
> ## RC1. Patch- vs record-level privacy and when to use each
> The choice of neighboring relation is **indeed fundamental to DP**. This is analogous to why DP is usually analyzed at the record-level rather than the user- or database-level: coarser relations are always valid but can be overly pessimistic when sensitive content is constrained to a subset of the input. Patch-level DP is appropriate when this assumption holds; otherwise, standard record-level guarantees apply as a safe default. Additionally, in our method, patch-level privacy gracefully converges to record-level privacy as the private patch size nears the image size (see Figure 4).
>
> We already motivate when patch-level privacy is appropriate in the introduction (Section 1, paragraph 4). **Based on your feedback, we have added a discussion clarifying this relationship and when each is appropriate (see end of Section 4.1, blue text, and the new Appendix A).** Thank you for pointing this out!
>
> ## RC2.1. Assumptions about patch size
>
> Thank you for the great question!
>
> Importantly, there is no catastrophic failure of privacy under misspecification; instead, privacy degrades gracefully.
>
> If the assumed patch is larger than the true sensitive region, guarantees remain valid but conservative. Privacy decays following the group privacy property (see, e.g., [1]), but the privacy profile for any patch size can also be drawn tightly via our analysis.
>
> If the assumed patch is smaller than reality, the stated $\varepsilon$ is optimistic; however, this is a general limitation in DP, analogous to assuming record-level privacy when user-level privacy is required.
>
> In either case, assuming the entire image is sensitive, it recovers standard DP-SGD, which always provides a valid guarantee. Figure 4 illustrates this directly.
>
>
> Crucially, our **analysis is decoupled from the mechanism**: we can compute guarantees for any combination of crop size and private patch size without modifying training.
>
> We have updated Section 6, paragraph 3, to clarify this.
>
> ## RC2.2. Multiple sensitive regions
>
> This follows similar reasoning to the patch size question above. For multiple disjoint sensitive regions, one could (1) use the group privacy property of DP [1] or (2) assume the union of patches as a single private composite region and apply our tight analysis directly.
>
> In either case, there is no catastrophic failure; the worst-case limit of patches covering the entire image recovers standard DP-SGD.
>
> Based on your feedback, we have added "Generalization to Multiple Patches" in Section 4.4 to clarify this more explicitly (see blue text).
>
>
> ### References:
>
> [1] Vadhan, Salil P.. “The Complexity of Differential Privacy.” Tutorials on the Foundations of Cryptography, 2017.

---

> > ### Author Response · Authors · 2026-01-02
> > **Rebuttal by Authors - part 2**
> >
> > ## RC4.1. Definition 3 allows arbitrary changes within the region
> >
> > Yes, your interpretation is correct! Outside $R$, images must be identical; inside $R$, any modification, including a single-pixel change, constitutes adjacency.
> >
> > This mirrors standard record-level DP-SGD, where changing a single attribute of a record is treated identically to changing all attributes. This is because the only available information about the model's gradients is their clipping norm. Thus, we have to make the worst-case assumption that the gradient changes arbitrarily within said norm, even when a single input pixel changes. We cannot assume that small input changes yield small gradient changes.
> >
> > Our work can be interpreted as refining the worst-case assumption for DP-SGD: instead of assuming the entire image may change (record-level), we assume only a bounded region may change (patch-level). When the domain structure supports this assumption, tighter guarantees follow.
> >
> > We have clarified this in the manuscript by adding clarification to Section 4.1, directly after Definition 3 (see blue text).
> >
> > ## RC4.2. Clarify if the entire region R is privatized
> >
> > The entire region $R$ **is indeed always protected** by the DP guarantee. What Figure 1 illustrates is the source of privacy amplification: when the private patch is excluded from a crop (Figure 1, middle), it cannot influence the gradient. If even a subsection of the private patch is included (Figure 1, right), the gradient is affected by the private information. This stochasticity amplifies privacy.
> >
> > The key point: we treat any overlap as full inclusion (binary treatment). Even if only one pixel of R appears in the crop, we conservatively assume the entire R influences the gradient. This is why Figure 1 (right) shows R fully highlighted, even when there is a partial intersection.
> >
> > We added a clarifying sentence to the Figure 1 caption (see blue text). We also now discuss this binary inclusion model in depth in Section 6, paragraph 2.
> >
> > ## R5. Choices of $\delta$
> >
> > Thank you for pointing this out!
> >
> > We have updated the manuscript and all experiments now consistently use $\delta = 1/\text{epoch size}$.
> >
> > We have updated Section 5.2 in our main text, and Appendix C.4, D.2, D.3 accordingly (see blue text).
> >
> > ## R6. Discuss high error bars and reproducibility concerns
> > As described in Appendix C.4, we used a single fixed hyperparameter configuration across all experiments to ensure fair comparison. This configuration was tuned to maximize utility in DP setting, rather than to reduce variance. This, together with the known sensitivity of DP-SGD to hyperparameters [1, 2, 3, 4], explains the larger error bars for DeepLabV3+ on Cityscapes.
> >
> > **To fully address this concern, we now provide per-seed results for all model-dataset combinations in Appendix C.5.** These plots show that patch-level sampling consistently outperforms the baseline across seeds and privacy levels, with only one exception: a single seed at a single $\varepsilon$ value where the baseline briefly exceeds our method.
> >
> > Following your suggestion, we have also added a discussion of this to Section 5.3 (see blue text).
> >
> > ## R7. Stating $\delta$ and patch size in captions
> >
> > We have updated all captions of privacy-utility figures (see blue text in updated/new Figures 5, 6, 7, 8, 9, 10, 11, 12, 13, 14, 15). Thank you for pointing this out.
> >
> > ## Broader impact. Potential for misuse
> >
> > Thank you for raising this point. Choosing the appropriate granularity for a given threat model is standard practice in differential privacy. As mentioned in our previous responses, this is analogous to the choice of record-level privacy over user-level privacy. It is a modeling decision, not a flaw.
> >
> > When the no sound assumption of a small private patch can be made, practitioners should conservatively assume larger patch sizes, or in the limit, assume the entire image is sensitive. As mentioned previously, our method naturally recovers standard record-level DP-SGD in this case.
> >
> > We have added a note on responsible use to Section 6, third paragraph, to help practitioners avoid misuse.
> >
> > ### References:
> >
> > [1] Ponomareva, Natalia, et al. "How to dp-fy ml: A practical guide to machine learning with differential privacy." Journal of Artificial Intelligence Research 77 (2023): 1113-1201.
> >
> > [2] Sander, Tom, Pierre Stock, and Alexandre Sablayrolles. "Tan without a burn: Scaling laws of dp-sgd." International Conference on Machine Learning. PMLR, 2023.
> >
> > [3] De, Soham, et al. "Unlocking high-accuracy differentially private image classification through scale." arXiv preprint, 2022.
> >
> > [4] McKenna, Ryan, et al. "Scaling laws for differentially private language models." arXiv preprint, 2025.

---

### Review · Reviewer_htuJ · 2025-12-09

**Summary Of Contributions:**

This paper introduces patch-level differential privacy for vision models, redefining the neighboring relation to allow localized substitutions within an image. The work shows that the standard technique of random cropping acts as a free privacy amplification mechanism. The authors derive tight patch-inclusion probabilities, showing how this composes with minibatch subsampling, and demonstrate meaningful privacy-utility improvements on semantic segmentation tasks (DeepLabV3+, PSPNet; Cityscapes, A2D2).

Strength:

-- The key insight that random cropping inherently introduces a second, spatial sampling process is elegant and interesting. The paper formalizes something practitioners implicitly rely on yet never quantified: sensitive content may simply not appear in a crop, reducing exposure probability.

-- The derivation of patch-inclusion probability is correct and intuitive. The privacy amplification result (Theorem 2) generalizes standard subsampling bounds. The proofs are clean and the authors show tightness for DP-SGD via a constructive worst-case example. The connection to dominating pairs and PLD accounting is also thorough and interesting.

-- The overall idea is intuitive and simply a new privacy definition more aligned with actual risk scenarios.

Weakness:

-- The approach assumes a known fixed private patch (e.g., 10×10). In real datasets, sensitive regions (like faces or license plates) vary in position, size and shape. This limits deployment unless conservative over-estimates are used. The paper acknowledges this but does not explore adaptive or detection-based patch modeling.

-- A crop is treated as either “includes sensitive region” or “does not”. But in practice, even partial inclusion may leak varying amounts of information. A more graded, overlap-aware formulation could yield tighter (less conservative) privacy bounds.

-- The amplification benefits require high resolution images, random cropping with substantial downscaling and spatially localized privacy-critical information. Tasks without cropping (e.g., CIFAR-10, MNIST, small-resolution datasets) do not benefit. This limit is mentioned but deserves sharper discussion on scope.

-- While done for fairness, some DP-SGD baselines may be under-tuned. For example, DeepLabV3+ on A2D2 shows instability. Larger crop sizes or tuned LR schedules could shift baselines. The improvements remain credible but could partly reflect suboptimal baselines.

-- All experiments assume a small, square patch at arbitrary location. Real sensitive patterns can be elongated (license plates), elliptical (faces) or distributed clusters. Some experiments or simulations with varied geometries would strengthen the claim of generality.

**Audience:**

Yes

**Audience Explanation:**

Yes. This work is clearly of interest to a segment of the TMLR audience. It introduces a simple yet impactful refinement to differential privacy accounting that leverages a ubiquitous vision operation (random cropping) to obtain stronger privacy guarantees “for free.” Researchers working on DP-SGD, privacy amplification, vision models and domain-aware privacy mechanisms would benefit from these findings, as the paper offers both a novel theoretical perspective and practical improvements without altering training pipelines.

**Broader Impact Concerns:**

Nil

**Claims And Evidence:**

Yes

**Claims Explanation:**

Yes, the claims are supported by clear and convincing evidence. The theoretical guarantees are rigorously derived, with tightness results that validate the amplification effect under patch-level substitutions. The empirical results further reinforce these claims. Privacy profiles consistently show stronger guarantees than minibatch subsampling and the privacy–utility experiments across two architectures and datasets demonstrate substantial, repeatable gains. While some assumptions, such as fixed patch size—could be clarified, the core arguments are accurate, well-justified and strongly backed by both theory and experiments.

**Requested Changes:**

-- The current analysis assumes a fixed and known private region size (e.g., 10×10). Please clearly articulate how practitioners should choose this parameter in real-world settings where sensitive regions vary significantly in size and shape (e.g., faces, license plates).

-- Add a discussion about conservative bounding vs. dataset-dependent estimation.

-- Provide clearer justification for the binary inclusion model. The analysis treats crop–patch interactions as a binary event (“intersects” vs “does not”).

-- Clarify the implications of ignoring fractional overlap.

-- Explain how conservative this approximation is, and whether it can substantially loosen the privacy guarantee. A brief note on what would be required to extend the analysis to overlap-aware weighting would improve rigor.

-- Strengthen discussion on scope and limitations of applicability. Patch-level amplification only helps when cropping is standard practice and when images are high resolution. Please expand the limitations section to explicitly note tasks that will not benefit (e.g., small-resolution datasets, non-cropping pipelines).

---

> ### Author Response · Authors · 2026-01-02
> **Rebuttal by Authors - part 1**
>
> Thank you for your detailed and constructive review. We appreciate your recognition of our key insight and clean theoretical derivations. Based on your valuable feedback, we have updated our manuscript. For your convenience, all changes are highlighted in blue, and a summary of important changes is provided in our global rebuttal comment. We address all your concerns below.
>
>
> ## RC1 & W1.1. Choosing the private patch size in practice.
> Indeed, in real datasets, sensitive regions vary in both size and shape across images. Practitioners should choose a patch that upper-bounds the largest sensitive region in the dataset. This requires domain knowledge, just like practitioners having to decide between record-level and user-level DP based on their threat model (see Section 4.1 and Appendix A for a discussion of patch-level and record-level privacy).
>
> Concretely, such bounds can be derived from physical constraints: a dashboard camera at a known height with a license plate of standard size implies a maximum number of pixels; similarly, a surveillance camera at a known elevation bounds how large a face can appear. With such knowledge, one can use tighter patches or arbitrary shapes (Section 4.4, "Generalization to Arbitrary Regions"), yielding stronger privacy guarantees. A potential direction for future work could be detection-based patch modeling, which we discuss in detail below.
>
> We now discuss this in our manuscript in Section 6, paragraph 3.
>
> ## RC2. Conservative bounding vs. dataset-dependent estimation.
> This connects to the core thesis of our work: stronger guarantees are possible when domain knowledge constrains the neighboring relation. Crucially, this is not a binary choice but a spectrum. The same principle applies to patch size (Figure 4), placement (Lemma 1 vs. Theorem 1), and shape (Section 4.4). More knowledge enables tighter bounds, less knowledge requires looser ones, and the worst case (no knowledge) recovers standard DP-SGD exactly. Importantly, when in doubt, practitioners should conservatively assume a larger patch size.
>
> We have added a discussion of this to Section 6, paragraph 3 of our manuscript.
>
> ## W1.2. Detection-based patch modeling.
> One potential future direction for dataset-dependent estimation is using domain-specific detectors (e.g., face or license plate detection) to estimate patch statistics. However, this must be treated carefully:
>
> - The detector itself must be DP-trained, internal, or trained on public data with a matching distribution
>
> - False negatives (missed detections) would violate privacy guarantees
>
> This is a promising idea for future work, but our focus is on providing guarantees that hold under worst-case assumptions. As you have written, we already mentioned this point in our manuscript, but we have now added a more detailed discussion to Section 6, paragraph 3.
>
> ## Binary inclusion model
>
> Several comments ask about our binary treatment of the crop-patch intersection. We address them together. **Crucially, this is a fundamental property of DP-SGD**, not a simplification we chose. Since the gradient is an arbitrary function of the input, we cannot assume that seeing 10% of a sensitive region leaks less private information than seeing 100%. Even a single visible pixel could produce an arbitrarily different gradient. This is the same reason standard record-level DP-SGD treats changing a single attribute inside a record identically to changing all attributes.
>
> ### W2 & RC3. Justify binary inclusion.
>
> Given the above, binary treatment is necessary for sound privacy guarantees without additional assumptions about model behavior.
>
> ### RC4 & RC5.1. Clarify implications of ignoring fractional overlap. How conservative is this?
>
> We potentially overestimate privacy loss when partial overlap truly leaks less private information than full inclusion. However, without assumptions about the loss landscape, no tighter analysis is possible while remaining sound.
>
> ### RC5.2. What would overlap-aware analysis require?
>
> Tighter guarantees could be achieved with additional knowledge about the model. For example, if we know the gradients are Lipschitz continuous and can bound how much each pixel changes. Similar non-uniform sensitivity analyses exist for generic DP-SGD (see, e.g., [1, 2]). However, **our focus is on providing a general tool for DP-SGD that works for arbitrary models without such assumptions.**
>
> We agree that incorporating such model-specific knowledge could be a promising direction for future work. We have added this discussion to Section 6 , paragraph 2 (see blue text). Thank you for the suggestion!
>
> References:
>
> [1] Das, Rudrajit et al. “Beyond Uniform Lipschitz Condition in Differentially Private Optimization.” ICML, 2022.
>
> [2] Béthune, Louis, et al. "DP-SGD Without Clipping: The Lipschitz Neural Network Way." ICLR, 2024.

---

> > ### Author Response · Authors · 2026-01-02
> > **Rebuttal by Authors - part 2**
> >
> > ## RC6. Strengthen the discussion on the scope and limitations of applicability
> >
> > We agree this should be strengthened. Our method provides amplification when:
> >
> > - Random cropping is part of the pipeline (regardless of task: segmentation, detection, classification, etc.)
> >
> > - Images are high resolution relative to sensitive content
> >
> > - Private information is spatially localized
> >
> > Our method does *not* help when:
> > - No cropping is used (e.g., small datasets like CIFAR-10, MNIST)
> > - No exclusion probability (crop size $\simeq$ image size, or global privacy risk)
> >
> >
> > Many important real-world applications meet our requirements: autonomous driving, medical imaging, surveillance, and satellite imagery.
> >
> > We have expanded Section 6, paragraph 4, to explicitly note these conditions.
> >
> >
> > ## W5. Experiments with varied geometries
> >
> > Thank you for this suggestion. As you have also acknowledged, our theory already supports arbitrary shapes (see "Generalization to Arbitrary Regions" in Section 4.4). While we derived closed-form expressions for rectangular patches (Lemma 1), the inclusion probability can be computed for any shape by iterating over all crop positions.
> >
> > To strengthen the empirical support, we have added privacy profiles for varied geometries:
> >
> > - Rectangles
> > - Circular regions
> > - Random geometry (cluster of 2 circles)
> >
> > See newly added Figure 21 and discussion in Appendix D.1.6 (see blue text). These results confirm that our framework generalizes beyond square patches as the theory predicts.

---

### Review · Reviewer_8qgn · 2025-12-19

**Summary Of Contributions:**

This paper proposes a domain-aware refinement of differential privacy analysis for vision models trained with DP-SGD by showing that random cropping, a data augmentation technique, provides additional privacy amplification when sensitive information is spatially localized. The authors introduce a patch-level neighboring relation in which two datasets differ only within a bounded region of a single image, reflecting privacy risks such as faces or license plates and avoiding overly conservative image-level assumptions. Under this notion of privacy, random cropping is formalized as a stochastic mechanism that probabilistically excludes the private patch from the model’s input. The paper derives tight theoretical bounds showing that this patch-level randomness composes naturally with minibatch subsampling, resulting in a reduced effective sampling rate and stronger privacy guarantees for DP-SGD. The amplification is obtained without modifying the training algorithm, model architecture, or adding computational overhead, making it a drop-in improvement in privacy accounting. Experiments on semantic segmentation benchmarks further demonstrate improved privacy–utility trade-offs, validating the practical relevance of the proposed analysis.

**Audience:**

Yes

**Audience Explanation:**

The paper focuses on domain-aware refinement of differential privacy analysis and adopts random cropping as a solution, which may be of interest to a subset of the TMLR audience.

**Broader Impact Concerns:**

The paper does not have broader impact concerns.

**Claims And Evidence:**

No

**Claims Explanation:**

The paper presents a theoretical analysis demonstrating that patch-level randomness introduces additional privacy guarantees, and provides empirical results illustrating the effects of different sampling strategies and patch sizes on privacy, as well as the resulting privacy–utility trade-offs under patch-level subsampling. However, several aspects require further clarification or justification:

1. In Section 4.2, the criteria for selecting the crop dimensions $H_C$ and $W_C$ are unclear. In practice, if either $H_C$ or $W_C$ is too small, the crop may fail to preserve any private information, which could undermine the intended privacy analysis.

2. The patch inclusion probability is defined as the probability that a randomly sampled crop intersects with region R. It is unclear why this definition is adopted instead of considering the probability that a crop fully covers region R, given that the private information is removed only when the entire region R is substituted.

3. The role and interpretation of the parameter $\alpha$ in Theorem 2 are not clearly explained and should be explicitly defined.

4. Sections 5.1 and 5.2 restrict the analysis to rectangular crops and patches. However, as noted by the authors, patches may also take the form of irregular blobs. It would be helpful to clarify whether and how the proposed analysis extends to such non-rectangular patch shapes.

5. Finally, it would be nice to include a comparison of the privacy–utility trade-offs with the Gaussian data augmentation method proposed by Schuchardt et al. (2025), evaluated under the same experimental settings.

**Requested Changes:**

Besides the points raised above, the following additional revisions are requested:

1. The notation in Definition 3 requires further clarification. Specifically, x(s, t) should be defined before it is used. The region R is described as rectangular, whereas [H] x [W] appears to denote an arbitrary region; additional constraints should be specified to resolve this inconsistency. Moreover, the notation used for a dataset x and an individual sample x_i is potentially confusing; using X to denote the dataset would improve clarity.

2, The experimental evaluation would be strengthened by including additional datasets beyond the segmentation setting.

3. In practice, privacy budgets epsilon are typically small. It would therefore be beneficial to include more data points corresponding to smaller values of $\epsilon$ (for example, $\epsilon < 10$) in Figure 5.

---

> ### Author Response · Authors · 2026-01-02
> **Rebuttal by Authors - part 1**
>
> Thank you for your review and thoughtful questions. Based on your valuable feedback, we have updated our manuscript. For your convenience, all changes are highlighted in blue, and a summary of these changes is provided in our global rebuttal comment. We address all your concerns below.
>
> ## W1. Selection of crop sizes
>
> Thank you for the question.
>
> Our analysis works for **any** crop size. As long as crop size is non-zero, there is always a non-zero probability of intersection between the crop and private patch, since crop origins are sampled uniformly at random. Smaller crops yield lower intersection probability, resulting in **stronger** privacy guarantees. Larger crops increase intersection probability and reduce privacy amplification. This can also be seen in Equation 2 in Section 4.3.
>
> This is illustrated in Figure 3: as crop size increases, $\varepsilon$ increases (weaker privacy), eventually saturating at the standard DP-SGD baseline when the intersection probability reaches 1.
>
> Our framework provides the privacy analysis for any choice. In practice, crop size is a hyperparameter that affects both utility and privacy, which should be tuned.
>
> ### W2. Definition of patch inclusion probability
> Thank you for the question. We believe there may be a small misunderstanding: the crop region $C_{u,v}$ is the part of the image that is *kept and passed to the model*, not the part that is removed. Given this, we use intersection (rather than full coverage) because: **if any part of region $R$ appears in the crop, it can influence the gradient**, hence leak information.
>
> Since the gradient is an arbitrary function of the input, we cannot assume that seeing part of $R$ leaks less information than seeing all of $R$; even a single visible pixel could produce an arbitrarily different gradient. This is the reasoning behind our binary treatment of the crop-patch interaction (see Section 6, paragraph 2 for further discussion of the binary inclusion model).
>
> Thanks to your feedback, we have further highlighted this in Section 4.3 (see blue text).
>
> ## W3. Role of alpha in Theorem 2
>
> Thank you for pointing this out. The parameter $\alpha$ relates to $\varepsilon$ via $\alpha = e^\varepsilon$, as shown in Proposition 1, where the hockey-stick divergence is first defined as $H_\alpha$, and then $(\varepsilon, \delta)$-DP is defined as $H_{e^\varepsilon}(M(x) \| M(x')) \leq \delta$. This notation follows standard DP literature (see, e.g., [1] or [2]). Given this relation, the role of $\varepsilon$, and therefore of $\alpha$, is discussed in depth and has been experimented on.
>
>
> We have tried to point to this relationship before Proposition 5: "Proposition 3 provides a simple and intuitive privacy amplification bound for $\varepsilon > 0$ (i.e., $\alpha > 1$)...". But to make the relation of $\alpha$ to $\varepsilon$ more explicit, we have added a brief clarification after Proposition 1 (see blue text).
>
> ## W4. Clarify how to extend to non-rectangular private patches
> Thank you for this suggestion. Our theory already supports arbitrary shapes, as discussed in "Generalization to Arbitrary Regions" in Section 4.4. We derived closed-form expressions for rectangular patches (Lemma 1), but for other shapes, the inclusion probability can be computed by iterating over all crop positions.
>
> Following your suggestion, we have added privacy profiles for varied geometries to strengthen the empirical support:
>
> - Rectangles
> - Circular regions
> - Random blobs (cluster of 2 circles)
>
> These results are presented in the newly added Figure 21 and Appendix D.1.6 (see blue text), confirming that our framework generalizes beyond square patches as the theory predicts.
>
> ### References
>
> [1] Barthe, Gilles and Federico Olmedo. “Beyond Differential Privacy: Composition Theorems and Relational Logic for f-divergences between Probabilistic Programs.” International Colloquium on Automata, Languages and Programming, 2013.
>
> [2] Zhu, Yuqing et al. “Optimal Accounting of Differential Privacy via Characteristic Function.” AISTATS, 2021.

---

> > ### Author Response · Authors · 2026-01-02
> > **Rebuttal by Authors - part 2**
> >
> > ## W5. Comparison with Gaussian data augmentation (Schuchardt et al., 2025)
> > Thank you for the suggestion.
> >
> > We have added a comparison with Gaussian data augmentation as a privacy amplification mechanism, following the analysis of Schuchardt et al. (2025) [1].
> > In this setup, we add Gaussian noise to images during training and account for the resulting amplification by multiplying the subsampling rate by the worst-case total variation distance between the Gaussian distributions (see their Theorem 4.5). We calibrate the noise level to match the same target $\varepsilon$ values as our other experiments.
> >
> > Results for DeepLabV3+ on Cityscapes are shown in Figure 11. Keeping all other parameters the same, Gaussian noise augmentation performs  worse than both patch-level and standard minibatch subsampling. Our patch-level analysis achieves better privacy-utility tradeoffs by leveraging the inherent randomness of cropping, which is already part of standard training pipelines and does not introduce additional perturbations to the input. We discuss this further in Appendix C.7.
> >
> > Due to time and computational constraints, we currently report results for one seed for DeepLabV3+ on Cityscapes. We will add the remaining seeds and model-dataset configurations for the camera-ready version.
> >
> >
> > ## RC1. Notation clarifications in Definition 3
> >
> > Thank you for these suggestions.
> >
> > $x_i(s,t)$: This denotes the pixel value at coordinates $(s,t)$ in image $x_i$. We have added this clarification to Definition 3.
> >
> > $[H] \times [W]$ and rectangular R: We have replaced $[H] \times [W]$ with $[H_I] \times [W_I]$ for consistency. The rectangular structure of $R$ is now stated explicitly in the text and formalized in Lemma 1, which defines $R$ by its bottom-left corner and dimensions.
> >
> > $x$ vs $\mathcal{X}$: This follows standard DP notation (e.g., [2]) where $\mathcal{X}$ denotes a set of datasets, $x$ a dataset. We have now ensured the distinction between $\mathcal{X}$, $x$, and $x_i$ is clear in the text (see updated Definition 3, blue text).
> >
> >
> > ## RC2. Add non-segmentation experiments
> >
> > The privacy amplification we propose is **task-agnostic**: it improves DP-SGD accounting whenever random cropping is used, regardless of the learning task. Our analysis yields strictly stronger privacy guarantees when the crop can exclude the private patch (see Figures 2 and 16-21), and recovers the standard DP-SGD exactly in the worst case.
> >
> > To validate this empirically, we have added classification experiments with ResNet-18 and VGG-11 on DTD and MNIST (see updated Section 5.3 and Appendix C.8, in blue).
> >
> > On DTD, the improved privacy-utility tradeoffs clearly generalize to classification (see Figure 12).
> >
> > On MNIST, cropping substantially harms utility, even without privacy noise ($\varepsilon = \infty$; see Table 1). This does not contradict our claims. Our method still provides stronger privacy guarantees, but these do not translate to better performance because cropping destroys semantic information before noise is even applied. We include MNIST results to explicitly show when the practical benefit is limited (see Figure 13), and have highlighted this in our limitations discussion (see Section 6, paragraph 4).
> >
> > ## RC3. Add smaller epsilon values
> >
> > Thank you for the suggestion. We have added $\varepsilon=1$ to our experiments (see updated Figures 5 and the new Figures 6, 7, 11, 12, 13, 14, 15).
> >
> > We note that  small epsilon values (e.g., $\varepsilon < 1$) are uncommon in practical ML settings, as discussed in the "How to DP-fy ML" guide [3]. Smaller values often result in models that fail to learn meaningfully due to the magnitude of noise, and thus do not have the same practical relevance.
> >
> > We have run $\varepsilon=1$ for DeepLabV3+ on Cityscapes (all seeds, Figures 5, 6), PSPNet on Cityscapes (1 seed, Figure 7), and all classification experiments (all seeds, Figures 11, 12, 13, 14, 15). Due to time and computation constraints, we will add the remaining seeds and A2D2 experiments for the camera-ready version.
> >
> >
> >
> > ### References
> >
> > [1] Schuchardt, Jan, et al. Privacy amplification by structured subsampling for deep differentially private time series forecasting. ICML, 2026.
> >
> > [2] Dwork, Cynthia, and Aaron Roth. "The Algorithmic Foundations of Differential Privacy." 2014.
> >
> > [3] Ponomareva, Natalia, et al. "How to dp-fy ml: A practical guide to machine learning with differential privacy." Journal of Artificial Intelligence Research 77 (2023): 1113-1201.

---

### Author Response · Authors · 2026-01-02
**General Rebuttal Infos by Authors**

We are very grateful for the thoughtful and constructive reviews. We are particularly encouraged by the recognition of several strengths in our work: the elegant insight that random cropping introduces spatial subsampling "for free" (Reviewer htuJ), the strong theoretical link between data augmentation and privacy amplification (Reviewer HPaG), and the clean derivation of tight privacy bounds (Reviewer htuJ). We are glad that our motivation was found to be clear and intuitive, and that the practical relevance of improved privacy-utility tradeoffs was appreciated.

We have addressed all reviewer concerns individually and updated our manuscript accordingly. All changes are marked in blue. Key additions include:

**New experiments:**
- Classification experiments on DTD and MNIST using ResNet-18 and VGG-11 (Section 5.3, Appendix C.8) [HPaG: W1, RC3; 8qgn: RC2]
- Privacy profiles for varied patch geometries (Appendix D.1.6) [htuJ: W5; 8qgn: W4]
- Comparison with Gaussian noise augmentation (Appendix C.7) [8qgn: W5]
- Results for $\varepsilon = 1$ across all experiments [8qgn: RC3]

**Clarifications and discussion:**
- Setting private patch size in practice (Section 6) [HPaG: RC 2.1, Broader impact; htuJ: RC1, W.1.1, RC2]
- Patch-level vs record-level privacy and when each is appropriate (Section 4.1, Section 6, Appendix A) [HPaG: RC1]
- Binary inclusion model (Section 4.3, Section 6) [htuJ: W2, RC3, RC4, RC5]
- Generalization to multiple patches and arbitrary shapes (Section 4.4, Appendix D.1.6.) [HPaG: RC2.2; htuJ: W5; 8qgn: W4]

**Corrections:**
- Unified $\delta$ notation throughout (Section 5.2; Appendix C.4, D.2, D.3) [HPaG: R5]
- Added $\delta$ and patch size to all figure captions [HPaG: R7]
- Clarified Definition 3 notation [8qgn: RC1]

We address each reviewer's concerns in detail below and look forward to a fruitful discussion.

---

### Decision · Action_Editor_UntU · 2026-02-01

**Recommendation:** Accept as is

**Audience:**

Yes

**Audience Explanation:**

The work is of clear interest to TMLR's audience, particularly those in computer vision and privacy-preserving machine learning.

**Claims And Evidence:**

Yes

**Claims Explanation:**

The claims are supported by convincing evidence following the revisions. Reviewers initially raised valid concerns regarding the binary inclusion model, the limitation of the theory to rectangular patches, and the restriction of experiments to semantic segmentation tasks.  The authors successfully addressed these by clarifying that the binary model is a necessary worst-case assumption for arbitrary gradients, expanding the theory and experiments to cover arbitrary patch shapes, and adding additional benchmarks and comparisons. Following these additions, all reviewers agreed that almost all their concerns are addressed and they are leaned toward acceptance.